# Calibration of temperature-dependent ocean microbial processes in the cGENIE.muffin (v0.9.13) Earth system model

Katherine A. Crichton[1*], Jamie D. Wilson[2], Andy Ridgwell[3], Paul N. Pearson[1].

[1] School of Earth and Ocean Sciences, Cardiff University, UK
[2] BRIDGE, School of Geographical Sciences, University of Bristol, Bristol, UK
[3] Department of Earth and Planetary Sciences, University of California, Riverside, CA 92521, USA
* now at School of Geography, University of Exeter, EX4 4RJ, UK

Correspondence to: Katherine A. Crichton (k.a.crichton@exeter.ac.uk)

**Abstract.** Temperature is a master parameter in the marine carbon cycle, exerting a critical control on the rate of biological transformation of a variety of solid and dissolved reactants and substrates. Although in the construction of numerical models of marine carbon cycling, temperature has been long-recognised as a key parameter in the production and export of organic matter at the ocean surface, its role in the ocean interior is much less frequently accounted for. There, bacteria (primarily) transform sinking particulate organic matter (POM) into its dissolved constituents and consume dissolved oxygen (and/or other electron acceptors

such as sulphate). The nutrients and carbon thereby released then become available for transport back to the surface, influencing biological productivity and atmospheric $p$CO$_2$, respectively. Given the substantial changes in ocean temperature occurring in the past, as well as in light of current anthropogenic warming, appropriately accounting for the role of temperature in marine carbon cycling may be critical to correctly projecting changes in ocean deoxygenation as well as the strength of feedbacks on atmospheric $p$CO$_2$.

Here we extend and calibrate a temperature-dependent representation of marine carbon cycling in the cGENIE.muffin Earth system model, intended for both past and future climate applications. In this, we combine a temperature-dependent remineralisation scheme for sinking organic matter with a biological export production scheme that also includes a dependence on ambient seawater temperature. Via a parameter ensemble, we jointly calibrate the two parameterisations by statistically contrasting model projected fields of nutrients, oxygen, and the stable carbon isotopic signature ($\delta^{13}$C) of dissolved inorganic carbon in the

ocean, with modern observations. We additionally explore the role of temperature in the creation and recycling of dissolved organic matter (DOM) and hence its impact on global carbon cycle dynamics.

We find that for the present-day, the temperature-dependent version shows as-good-as or better fit to data than the existing tuned non temperature-dependent version of the cGENIE.muffin. The main impact of accounting for temperature-dependent remineralisation of POM is in driving higher rates of remineralisation in warmer waters, in turn driving a more rapid return of

nutrients to the surface and thereby stimulating organic matter production. As a result, more POM is exported below 80m but on average reaches shallower depths in mid and low latitude warmer waters, as compared to the standard model. Conversely, at higher latitudes, colder water temperature reduces the rate of nutrient resupply to the surface and POM reaches greater depth on average

as a result of slower sub-surface rates of remineralisation. Further adding temperature-dependent DOM processes changes this overall picture only a little, with a slight weakening of export production at higher latitudes.

As an illustrative application of the new model configuration and calibration, we take the example of historical warming and briefly assess the implications for global carbon cycling of accounting for a more complete set of temperature-dependent processes in the ocean. We find that between pre-industrial (ca. 1700) and present (year 2010), in response to a simulated air temperature increase of 0.9°C and an associated projected mean ocean warming of 0.12°C (0.6°C in surface waters and 0.02°C in deep waters), a reduction in particulate organic carbon (POC) export at 80m of just 0.3% occurs (or 0.7% including a temperature-

dependent DOM response). However, due to this increased recycling nearer the surface, the efficiency of the transfer of carbon away from the surface (at 80m) to the deep ocean (at 1040m) is reduced by 5%. In contrast, with no assumed temperature-dependent processes impacting production or remineralisation of either POM or DOM, global POC export at 80m falls by 2.9% between the pre-industrial and present day as a consequence of ocean stratification and reduced nutrient resupply to the surface. Our analysis suggests that increased temperature-dependent nutrient recycling in the upper ocean has offset much of stratification-induced

restriction in its physical transport.

## 1 Introduction

The cycle of carbon through the ocean is dominated by the production, destruction, and transformation of both dissolved and particulate organic matter (DOM and POM, respectively) (Legendre et al., 2015; Heinze et al., 2015). The 'biological carbon pump' (Fig. 1) is central in this – acting through phytoplankton growth to remove carbon (and nutrients) from the surface and mixed layer

waters, and subsequently transferring it to the deep, principally via the sinking of POM (see: Hülse et al., 2017 for a review) and to a lesser extent, via the subduction of DOM. Export of POM out of the near-surface euphotic zone is principally controlled by photosynthesis rates (primary production) together with zooplankton grazing, respiration, and other food web processes (Steinberg and Landry, 2017; Mari et al., 2017). Of this export, only a fraction ultimately reaches the deep ocean as the initial export flux is filtered through a series of further processes and transformations involving feeding and remineralisation by microbes and other

biota, and further modulated by sinking speeds and composition of the sinking matter itself (Bach et al., 2016; Rosengard et al., 2015; Turner, 2015). At the ocean floor, organic matter undergoes further microbial degradation and transformation before the residual is buried in accumulating marine sediments. Removing carbon from surface waters and storing it for centuries (intermediate depths), millennia (deep ocean), or multi-millennia (sediments) exerts an important control on atmospheric $p\text{CO}_2$, which would otherwise be some 150ppm to 200ppm higher than present in the absence of any biological activity in the ocean (Parekh et al 2005,

Sarmiento and Gruber, 2006). Ocean circulation generally acts against the biological pump, geochemically homogenising the ocean interior and returning carbon (and nutrients) back to the surface. Surface-to-deep geochemical gradients and storage of carbon in the ocean is hence a function of the rate of ocean ventilation in conjunction with the rate of biological export of carbon from the surface and importantly, the rate at which the organic matter is remineralised in the ocean interior.

To a first order, export of carbon out of the mixed layer in a warming world will be reduced as a consequence of increased ocean stratification reducing nutrient re-supply to surface waters (Portner et al., 2014; Reusch and Boyd, 2013). However, the export of carbon is only one of the pertinent factors in marine carbon cycling. Also important is the 'transfer efficiency' of the biological pump – the fraction of exported carbon that reaches the inner ocean, or alternatively, the mean depth below the surface at which this carbon is remineralised, and dissolved inorganic carbon (DIC) returned to the ocean. A deeper mean remineralisation depth equates to a more transfer-efficient biological carbon pump. The sub-surface processes that affect the biological carbon pump efficiency are also temperature-dependent (Bendtsen et al., 2015; Turner 2015; Boscolo-Galazzo et al 2018), complicating the net response of the biological carbon pump and carbon sequestration in the ocean interior to changes in global warming.

Not all ocean biogeochemical models and associated global carbon cycle studies account for temperature controls on (and hence efficiency of) the biological carbon pump (some that do include: Kvale et al., 2019; Yamamoto et al., 2018; Cao and Zhang 2017; Laufkotter et al., 2016; Kvale et al., 2015; Segchneider and Bendtsen, 2013; Chikamoto et al. 2012; Moore et al. 2002). Hulse et al. (2017) presented an extensive review of how EMICs (Earth System Model of Intermediate Complexity) and box models treat ocean carbon cycle processes and summarised how inner ocean processes are less well constrained than surface processes in many models and how their treatment in models is much more variable. This is also the case for more complex ocean models, such as those participating in CMIP5 (Coupled Model Inter-comparison Project 5) and used to inform the recent IPCC assessment (Bopp et al.2013, Burd et al., 2016). However, temperature-dependency of inner ocean processes has been found to have an important impact on nutrient distribution and, therefore, on primary production (Tauscher and Oschlies, 2011) as well as biological pump efficiency (Laufkotter et al., 2017), arguing for the necessity of their inclusion in models.

All things being equal, higher ocean temperatures should drive a greater fraction of remineralisation to occur in the upper water column, facilitating increased carbon and nutrient return to the ocean surface. Higher surface temperature will also increase the ability of phytoplankton to take up and assimilate the resulting increased nutrient availability and hence re-export POM, although grazing pressure would also increase. However, higher temperatures also lower the solubility of $CO_2$ at the surface and hence loss to the atmosphere. Furthermore, for a geologically rapid and transient warming at the surface such as is currently occurring, increased ocean stratification and reduced physical transport of carbon and nutrients back to the surface will occur. The multiple conflicting influences of temperature mean that even the net sign of the feedback between greenhouse warming and ocean carbon cycling is uncertain (Yamamoto et al 2018). How carbon fixed at the surface is recycled via partitioning into DOM rather than exported as POM, and the rate at which DOM itself is recycled back into dissolved carbon and nutrients as temperatures rise adds another complicating layer of temperature response and feedback.

To help tease apart the varying influences of temperature on marine carbon cycling and atmospheric $CO_2$, we present and calibrate a temperature-dependent representation of the biological pump in the current 'muffin' release of the cGENIE EMIC (Earth system Model of Intermediate Complexity) (Cao et al. 2009) (and see statement on 'Code Availability'). Our calibrated configuration is intended for use in global biogeochemical cycling studies that require a fuller consideration of the role of temperature both in the geological past and the future. For completeness, we additionally develop and test a pair of parameterisations for temperature-dependency in the production and decay of DOM.

## 2 The cGENIE.muffin Earth system model framework (STND)

The basic framework of the cGENIE EMIC consists of a 3D frictional-geostrophic approximation ocean circulation model (Edwards and Marsh, 2005), coupled to a 2D dynamic-thermodynamic sea-ice model (Marsh et al., 2011). As per previous calibrations of ocean biogeochemical cycles (e.g. Ridgwell et al., 2007), we employ the ocean circulation and sea-ice model on a 36 x 36 equal area grid (10 degrees of longitude and uniform in the sine of latitude), and couple these with a 2D energy-moisture-balance atmosphere model (EMBM) (Marsh et al., 2011) (an alternative to this – a 3D atmospheric general circulation model (Holden et al., 2016) also exists, but is not employed in this study). For traceability, we employ a commonly-used physical configuration with 16 vertical levels in the ocean and a present-day bathymetry following Cao et al. (2009) and retain all physics parameter values and boundary conditions controlling the climate system. The representation of the ocean carbon and other biogeochemical cycles together with ocean-atmosphere gas exchange, unless otherwise noted, also follow Cao et al. (2009), and are summarised in more detail below. The various temperature-dependent parameterisations that we substitute for the equivalent non temperature-dependent processes in Cao et al. (2009) are described in full in this paper.

It is likely that both increased grazing pressure by zooplankton, and primary production by phytoplankton will have an impact on export production in a warmer world (Paul et al., 2015; Turner, 2015). However, in the simplified biologically induced export flux (Maier-Reimer, 1993) scheme (Fig. 2) that we apply in cGENIE, we cannot explicitly consider the impact of increased grazing pressure in the surface waters. Regardless, we are interested in the wider question of the interaction of (any) temperature-dependent community production (as export production), with temperature-dependent microbial remineralisation in the ocean interior, and its impact on the global ocean carbon cycle. We hence represent only the direct effect of temperature on large scale metabolic processes (plankton photosynthesis (growth) and microbial respiration). Other factors such as involving particle size distributions, particle density (Cram et al., 2018) and 'ballasting' (e.g. Wilson et al., 2012), and sinking speed (determined by particle characteristics) (McDonnell et al., 2015), are generally determined within the food web and may be considered to be of secondary importance in the context of the gross role of temperature in global carbon cycling. Recently, Boyd et al. (2019) defined additional particle pumps in the ocean, involving eddy-subduction, diel vertical migration, mesopelagic migration and seasonal lipid pumps. These processes are also outside the scope of this study, particularly in the absence of a sufficiently high resolution ocean circulation model component, the absence of (fully coupled GCM) ocean-atmosphere dynamics and inter-annual variability, and without an explicit ecosystem including multiple trophic layers and explicit zooplankton behaviours.

### 2.1 Standard, non-temperature-dependent model formulation (STND)

In the original version of the biological uptake scheme (Ridgwell et al., 2007), nutrients are taken out of the surface ocean layer according to several factors including light incidence, ice fraction, nutrient uptake limitation (Michaelis-Menten type), and a prescribed maximum uptake rate (Eq. 1). In this, $\Gamma$, the net nutrient uptake (mol $PO_4$ kg$^{-1}$ yr$^{-1}$) and hence net primary production in the surface ocean layer of the model, is described as:

$$\Gamma = u_0^{PO_4} \cdot \frac{PO_4}{PO_4 + K^{PO_4}} \cdot (1 - A_{ice}) \cdot \frac{I}{I_0} \quad \text{(1)}$$

where:

$u_0^{PO_4}$         maximum uptake rate (mol kg$^{-1}$ yr$^{-1}$)

$\frac{PO_4}{PO_4 + K^{PO_4}}$         nutrient limitation term

$PO_4$         local nutrient concentration (mol kg$^{-1}$)

$K^{PO_4}$         Michaelis Menten half saturation value (mol kg$^{-1}$)

$1 - A_{ice}$         ice free fraction of sea surface ($A_{ice}$ being the ice-covered fraction of a grid cell)

$\frac{I}{I_0}$         light limitation (based on incidence angle) term

Here, a maximum uptake rate, (maximum rate of conversion of dissolved PO$_4$, phosphate, into organic matter by phytoplankton) is
prescribed, and is assigned a calibrated value of 9.0 x10$^{-6}$ mol kg$^{-1}$ yr$^{-1}$ (Cao et al., 2009), while the calibrated Michaelis Menten half saturation value is 9.0 ×10$^{-7}$ mol kg$^{-1}$ (Cao et al., 2009).

Nutrient uptake is instantaneously converted into organic matter export, both particulate organic matter (POM) and a fraction as dissolved organic matter (DOM), in a ratio of 1:2 following Najjar et al. (2007), and this represents community production (see Fig. 2). This production encompasses the entire surface food web, including the action of primary producers (phytoplankton)
and the effect of consumers (e.g. grazers). In this export production model, dissolved inorganic carbon (DIC) is taken out of solution in the surface layer at a molar ratio of 106:1 to PO$_4$ and O$_2$ at a ratio of - 138:1 with PO$_4$ (Redfield et al., 1963). POM is partitioned into two fractions, which conceptually are: labile (fraction 1, 'POM1'), and recalcitrant POM (fraction 2, 'POM2') (Ridgwell et al., 2007). POM sinks vertically out of the surface layer and settles with a given velocity (here: 125 m day$^{-1}$). POM is remineralised throughout the water column using a prescribed remineralisation 'curve' reflecting the decay of POM as it sinks, first using dissolved
oxygen and then sulphate. The prescribed remineralisation 'curve' of relative sinking flux vs. depth (e.g. see: Hülse et al., 2017) is always adhered to (Eq. 2a for POM1, Eq. 2b for POM2). In the sinking curve, the relative flux at each layer (z) is calculated according to an exponential decay function (Ridgwell et al., 2007).

$$F_z^{POM1} = F_{z=h_e}^{POM1} \cdot \left( (1 - r^{POM}) + r^{POM} \cdot exp\left( \frac{z_{h_e} - z}{l^{POM1}} \right) \right) \quad \text{(2a)}$$

$$F_z^{POM2} = F_{z=h_e}^{POM2} \cdot \left( (r^{POM}) + r^{POM} \cdot exp\left( \frac{z_{h_e} - z}{l^{POM2}} \right) \right) \quad \text{(2b)}$$

where:

$F_{z=h_e}^{POM}$         POM exported out of the surface layer (at 80m)

$l^{POM}$         length-scale (556m for POM1; 1x10$^6$m for POM2 – effectively infinite and hence no water column decay)

$r^{POM}$          initial proportion of POM into fraction 2 (0.055)


Any POM not remineralised within the water column is instead remineralised at the ocean floor – a 'reflective' boundary condition assumption (see Hulse et al. (2017) for discussion). DOM is degraded with a lifetime of 0.5 years following Najjar et al. (2007).

## 2.2 Temperature-dependent processes (TDEP)

In an alternative representation of biological export production in the model, a temperature-dependent growth rate limiter is applied
to a characteristic time-scale of ambient nutrient depletion (Eq. 3). A similar scheme (but without the addition of temperature-dependent remineralisation in the ocean) has previously been applied by Meyer et al. (2016) for $PO_4$-only uptake, and for 2 nutrients ($PO_4$ and $NO_3$) by Monteiro et al. (2012). In this, net nutrient uptake (mol $PO_4$ kg$^{-1}$ yr$^{-1}$) is:

$$\Gamma = V_{max} \cdot \frac{PO_4}{PO_4 + K^{PO_4}} \cdot (1 - A_{ice}) \cdot \frac{I}{I_0} \cdot \gamma^T \cdot PO_4 \tag{3}$$

where:

$\gamma^T$                    temperature growth limitation term (see below)

$V_{max}$                    maximum net depletion rate multiplier (yr$^{-1}$)

$PO_4$                    local $PO_4$ concentration (mol kg$^{-1}$)

$\frac{PO_4}{PO_4 + K^{PO_4}}$          nutrient limitation term

$K^{PO_4}$                    Michaelis Menten half saturation value (mol kg$^{-1}$)

$1 - A_{ice}$          ice-free fraction of cell

$\frac{I}{I_0}$                    light limitation (based on incidence angle) term

Temperature growth limitation is represented by the Arrhenius equation, where T is local temperature (Eq. 4).

$$\gamma^T = ae^{(T/b)} \tag{4}$$

180          This is the "Eppley curve", commonly applied to model metabolic response to temperature change (Table 1). An improved fitted curve was proposed by Bissinger et al. (2008) (the LPD or Liverpool Plankton Database curve), with both being based on fitting the model to data from empirical studies. The largest difference between the Bissinger curve and the Eppley curve is the value of $a$ (Eq. 4, Table 1). It makes little difference which curve we use because we calibrate $V_{max}$ (Eq. 3) which is also a multiplier for the temperature growth limitation term (in Eq. 4). We use the original Eppley et al. (1972) values for $a$ (0.59) and $b$ (75.80), and
as per Monteiro et al. (2012). Both the Eppley and Bissinger curves gives a $Q_{10}$ value (where $Q_{10}$ is the increase in the rate of the metabolic process with a 10°C increase in temperature) for nutrient uptake as 1.88 (Bissinger et al. 2008).

To calculate the remineralisation rate of POM (mol yr$^{-1}$), an Arrhenius-type equation is applied (as in John et al., 2014) (Eq. 5).

$$k(T) = \beta_{POMn} e^{\left(-E_a/RT\right)}$$

(5)

where:

$E_a$    Activation energy (J mol$^{-1}$)

$R$    gas constant (J K$^{-1}$ mol$^{-1}$)

$T$    absolute temperature (K)

$\beta_{POMn}$  rate constant for POM remineralisation (yr$^{-1}$) as $T$ approaches infinity (for each POM fraction)


This rate is calculated in each ocean layer (and model grid point) according to the local temperature, POM flux, and for each of the two POM fractions. For both fractions, sinking rate is assumed to be 125 m day$^{-1}$ (Ridgwell, 2001), so for cGENIE's non-uniform ocean depths, the fractional loss of POM due to remineralisation in each layer ($z$) is as Eq. 6.

$$\Delta F_z^{POMn} = k(T)_z^{POMn} . \Delta t(z)$$

(6)

where $n$ denotes POM fraction (either labile (1) or recalcitrant (2) - distinguished as these have different $k(T)$ values), $\Delta t(z)$ is the time that sinking particles on average spend in layer $z$.

We will describe and evaluate the impact of processes accounting for the influence of temperature on both the production
(ratioed to POM) and degradation of DOM, as part of the Discussion section.

**3 Model tuning methodology**

In previous published applications of the cGENIE model, either a temperature dependence in calculating export (but not ocean interior mineralisation) (e.g. Meyer et al. 2016, Monteiro et al. 2012) OR a temperature dependence in remineralisation (but not biological productivity) (John et al., 2014) have been utilised in addressing varying paleo questions. More commonly, neither have
been utilised (e.g. Ridgwell and Schmidt, 2010). Here, we explore both temperature dependencies together (Table 2). Although in John et al. (2014) the temperature-dependent remineralisation scheme was calibrated to approximate the global mean POM decay profile in the default model (i.e. Ridgwell et al., 2007) under pre-industrial boundary conditions, here we adopt a more formal re-tuning against observations of the primary scaling factors in each scheme.

We identify three primary parameters requiring joint re-tuning: 1, the maximum nutrient uptake rate $V_{max}$ (Eq. 3) that scales
export production. 2, the activation energy, $E_a(1)$ (Eq.5) (the minimum energy required for the transformation of organic carbon into inorganic carbon through respiration processes for the remineralisation of labile POC1, Particulate Organic Carbon type 1)

where the labile POC1 dominates the export from the surface. 3, the fraction of recalcitrant POC2 (denoted as *rPOM* Eq.2a and Eq.2b) formed at the surface that reaches the very deep ocean.

The $V_{max}$ range was chosen by running a series of test simulations (a 10k year spin up with pre-industrial boundary conditions followed by the historical transient simulation forced with $CO_2$ data to the present day) varying $V_{max}$ while retaining the temperature-dependent remineralisation scheme adopted in John et al. (2014). From these results we selected a range of values for the multiplier $V_{max}$ that gave a reasonable agreement with $PO_4$ and $O_2$ distributions; these $V_{max}$ values are 4, 7 and 10. For the initial fraction of POC2, we took the John et al. (2014) value (with POC2 fraction of 0.008), and applied a range that encompassed 25% to 400% around that value. For the $E_a(1)$ (Eq.5) setting, John et al. (2014) used a value of 55 kJ/mol, the median of a range of 50 to 60 kJ/mol for labile POC identified in Arndt et al. (2013). We test a series of values for $E_a(1)$ within the range (Table 3). Our ensemble hence consisted of 3 different choices for $V_{max}$, 3 different choices for initial fraction of POC2, and 5 different choices for $E_a(1)$, for a total of $3\times3\times5 = 45$ different parameter combinations and hence model ensemble member experiments. Values for the two rate constants, $\beta_{POMn}$ (Eq 5, for POC1 at $9\times10^{11}$ yr$^{-1}$, for POC2 at $1\times10^{14}$ yr$^{-1}$) that were calibrated for the modern ocean and the sinking speed of 125 m day$^{-1}$ in John et al. (2014) are retained.

Each of the 45 experiments in the ensemble are spun-up for ten thousand years under pre-industrial boundary conditions: atmospheric $pCO_2$ is restored to 280 ppm and with a $\delta^{13}CO_2$ isotopic value of -6.5 ‰. To contrast the model ensemble members with observations, model-data comparison based on a simulated pre-industrial steady-state creates a mis-match because global datasets are based on modern (i.e. the past few decades) oceanographic observations, where (especially shallow) distributions of nutrients and oxygen have already been impacted by historical warming. Therefore, following on from each respective spin-up, each model ensemble member is then forced from year 1700 to 2010 in a transient simulation with atmospheric composition conforming to the observed mean annual trend in $CO_2$. Direct atmospheric measurements and ice core data has shown that atmospheric $\delta^{13}CO_2$ has dropped with increasing $CO_2$ due to fossil fuel emissions (that have a characteristic low $\delta^{13}C$) known as the Suess effect (Keeling, 1979; Rubino et al., 2013). This affects ocean $\delta^{13}C$ in a non-uniform manner – impacting (in general) nearer-surface waters more strongly due to ocean physics and circulation patterns. We hence additionally force atmospheric composition in the transient simulations with declining $\delta^{13}CO_2$ (Francey et al., 1999).

**3.1 Model-data comparison method**

For the model-data comparison, World Ocean Atlas 2009 (WOA 2009, Levitus et al. 2010) 5° gridded climatological data for phosphate ($PO_4$) and dissolved oxygen ($O_2$) was interpolated to a 10°x10° grid (with a simple linear upscaling), and then to the cGENIE model depth scale by averaging over the data depth points that most closely correspond to the cGENIE ocean model depth layer distribution. This depth rescaling produced a global mean depth-uncertainty of 2.2% (of the targeted cGENIE depth). The depth rescaling resulted in minor additional uncertainties of $\pm$ 0.01µmols l$^{-1}$ (at 1 standard deviation) for $PO_4$, and $\pm$ 0.02 µmols l$^{-1}$ for dissolved $O_2$ in addition to the error inherent in the gridded climatology product. For a direct comparison with the data, cGENIE model output from year 2010 of the transient experiments was also interpolated to 10°×10°, and converted to units of µmols l$^{-1}$ (from mol kg$^{-1}$ using modelled water density). Latitudes higher than 80° were neglected due to higher uncertainties in both data and

model outputs. For PO$_4$, we statistically compared the surface concentration, important for constraining nutrient uptake rates, as well as the global ocean distribution, which strongly reflects remineralisation and hence the strength and efficiency of the biological pump (plus ocean circulation). For dissolved O$_2$, we statistically compared model and data between 283m to 411m (cGENIE ocean level 4 centred at 346m) as an indication of the dissolved oxygen depletion caused by remineralisation near the bottom of the mixed layer and how well the model can represent this. As with phosphate, we additionally compared the model global ocean dissolved

oxygen distribution with data. Finally, given that we are utilizing model-derived temperature distributions in the ocean to project nutrient and oxygen concentrations (plus $\delta^{13}C$ distributions) which we then contrast with the respective observed data, we interpolated temperature data (producing an uncertainty of ±0.1°C) in the same way as O$_2$ and PO$_4$ so as to elucidate biogeochemical biases arising from model-data temperature mismatch.

To evaluate the model skill, we compared model to data standard deviation (SD), calculated the centred root mean square

(CRMSD) difference between model and data, and calculated model-data correlation, these parameters can then be presented on a Taylor plot. On this plot the general proximity of the model point to the data point indicates the goodness of fit, as well as a providing a visual comparison of how the changing parameters affect the model skill.

For assessing water column profiles, we defined a set of ocean regions as shown in Figure 3. These regions are similar to those used by Weber et al. (2016), but with some regions reduced in size or separated (Subtropical Pacific limited to South Pacific

and North Indian Ocean added). This was done so that within each region, ocean water characteristics (including temperature, nutrients, oxygen, salinity) as well as particle fluxes (as Weber et al. 2016) are broadly similar. We compare the model distribution of $\delta^{13}C$ of DIC with data from Schmittner et al. (2013) by grouping this data into regions (Fig. 3) and creating representative (mean with standard deviation) down-column profiles for $\delta^{13}C$ for visual comparison with model outputs in the matching region.

## 4 Model-data and model-model comparison

### 4.1 Tuning the temperature-dependent version – model vs. data

We first assess model skill in simulating the temperature distribution in the ocean, given its critical importance in the temperature-dependent calculations of metabolic processes (Fig. 4). We find a generally reasonable model fit to ocean temperature data in mid and low latitude near-surface waters, and in capturing the first order patterns in benthic temperatures. At high latitudes, cGENIE shows larger differences as compared with observations due to deficiencies in modelled ocean circulation and/or the surface climate

as simulated in the simple 2D EMBM. For instance, the temperature discrepancy throughout the water column in the North Atlantic may be due to an overly-strong AMOC (Atlantic Meridional Overturning Circulation) in the model that delivers too high a volume of warmer surface waters to depth. For the North Pacific, the model overestimation of near-surface temperatures by cGENIE likely reflects insufficient surface stratification; too-deep downwards mixing transporting too-warm surface waters to mid depths in this location.

The ensemble simulation results (year 2010 of the transient experiments) for the ocean geochemistry are shown in Fig. 5 on Taylor diagrams for their fit to observed distributions of dissolved PO$_4$ and O$_2$ in the ocean. Points in Fig. 5 are shaded according

to the $E_a(1)$ (Eq.5) value, which has the strongest control on $PO_4$ and $O_2$ distribution of the three variables. The overall best-fit to the data for the final TDEP setting was selected as $V_{max} = 10$, $E_a(1) = 54$ kJ mol$^{-1}$, initial fraction POC2 = 0.008, where the best-fit is determined as the setting with the combined overall lowest CRMSD for the $PO_4$ and $O_2$ distributions. The CRMSD for each variable is shown on Fig. 5 as the radial distance from the data point (the radial axis coloured green). In general the parameter value combinations with lowest CRMSD values corresponds with highest correlations in the model fit to data (Fig 5). The statistics for the STND and best fit TDEP are listed in Table 4.

## 4.2 Performance of the temperature-dependent model compared to the standard model

In this section we evaluate and compare the performance of the existing tuned, but non temperature-dependent biological scheme (STND), to the new tuned temperature-dependent scheme (TDEP).

Figure 6 shows cross-plots for surface $PO_4$ concentration (an important constraint on export production) for the best fit TDEP and the standard model for the selected ocean regions. The addition of temperature dependence in TDEP generally increases surface $PO_4$ concentrations and comes into better agreement with data than the standard model (STND). The high surface nutrient regions, Antarctic and the North Pacific, are lower than data in all model cases and this is likely due to the lack of iron limitation in this version of the model, as biological activity removes too much nutrient from the surface waters. However, in these regions, the temperature-dependent version shows a slightly better fit to data than does the standard model, as the colder surface water reduces nutrient uptake rates. In contrast, the lowest nutrient regions (e.g. some south Pacific and some east tropical Atlantic) have slightly higher $PO_4$ concentrations compared to data in TDEP.

Regional water-column profile model outputs for TDEP and STND are plotted against $PO_4$ data (Levitus et al. 2010) in Fig. 7. In both schemes, the model was tuned according to its fit to $PO_4$ (as well as $O_2$) and both TDEP and STND show a visually reasonable fit to data. Some differences can be seen between TDEP and STND in surface mid and low latitude waters, e.g. in the north Indian Ocean and eastern tropical Pacific, where nutrients are higher in TDEP in better agreement with data. In higher latitude waters, model-data mismatch may be more closely related to ocean circulation (as for temperature, Fig. 4) with STND and TDEP distribution very similar to each other in the Southern Ocean and North Pacific, but with and TDEP a better fit to data in the North Atlantic.

With the exception of the high southern latitudes (higher than ~ 60°S), the addition of temperature-dependent microbial processes generally increases surface nutrient concentrations (as shown in Fig. 8 a and b) as compared to the standard model. This is particularly apparent in the low-nutrient gyres, with up to 4-times higher $PO_4$ concentrations in TDEP as compared to in STND (Fig. 8b). In the deeper ocean (Fig 8 c and d), nutrient concentrations are reduced in the temperature-dependent version except for in the North Atlantic (where higher surface nutrients are delivered to the deep via the AMOC) and the high Southern latitudes (with slightly higher $PO_4$ than the standard model).

The distribution of dissolved oxygen also provides important information about the biological pump. Photosynthesis removes $CO_2$ from ocean waters and adds $O_2$ (where $CO_2$ and $O_2$ are also exchanged with the atmosphere at the surface ocean), while respiration does the opposite. As a general pattern, respiration progressively reduces dissolved oxygen concentrations down

the water column until a minimum is reached. Below that depth – the 'oxygen minimum zone' (OMZ) – ocean circulation reintroduces more oxygenated water masses from below. Dissolved oxygen concentrations then slightly increase again with further depth as the flux of organic matter and hence respiration declines, and oxygen is supplied through deep ocean circulation and via ventilation at the poles. In the Antarctic zone (Fig. 9), circulation patterns appear to dominate oxygen content (as STND and TDEP are very similar, but fairly dissimilar to data indicators). The North Pacific region also shows offsets between model and data for $O_2$, $PO_4$ and temperature, also suggesting deficiencies in simulated deep ocean circulation. In low latitude waters, TDEP shows a better fit to data between the surface and 500m than STND, suggesting that oxygen depletion rates due to respiration are better described here. Overall, the intensity of the OMZ in both STND and TDEP appear visually in reasonable agreement with data, although in low and mid latitudes warmer waters the OMZ occurs higher in the water column in TDEP than STND.

**4.3 Tracing Carbon-13**

In studying paleo climates and, in particular, past states of ocean circulation and carbon cycling, carbon-13 data is widely used as a tracer (Lynch-Stieglitz 2003). Surface waters have characteristic carbon-13 signatures with a pronounced mostly latitudinal gradient, so changes in $\delta^{13}C$ measured at any one location on the ocean floor may be at least partially attributed to changes in the strength and location of deep-water formation. The biological pump also affects the $\delta^{13}C$ signature of seawater. The process of photosynthesis fractionates the carbon that is exchanged (from the dissolved inorganic to the organic form); carbon-12 is preferentially taken up, leaving more carbon-13 in the surface waters (Schmittner et al. 2013). As summarized by Kirtland Turner and Ridgwell (2016), in the cGENIE model, fractionation between POC (and DOC) and $\delta^{13}C$ of $CO_2(aq)$ in cGENIE is a function of the $CO_2(aq)$ concentration and based on an approximation of the model of Rau et al. (1996) (Ridgwell, 2001). This gives rise to a spatial distribution in the $\delta^{13}C$ of exported organic carbon, with lower (more negative values) at higher latitudes, and higher (less negative) values towards the equator, primarily reflecting the temperature control on the concentration of $CO_2(aq)$ in surface waters. The mean flux-weighted $\delta^{13}C$ of POC is around -23‰ for the pre-industrial period, and around -26‰ by the year 2010 due to the impact of increasing $CO_2(aq)$ on organic carbon [13]C fractionation as well as the Suess effect. As POC is remineralised in the water column, low $\delta^{13}C$ carbon is released, modifying the ambient $\delta^{13}C$ of DIC. The $\delta^{13}C$ of the ocean interior then represents a balance between the input of light $\delta^{13}C$ via the biological pump, and the ingress of heavier $\delta^{13}C$ supplied in deep waters and ultimately sourced from high latitudes at the surface.

The regional mean and standard deviation of data $\delta^{13}C$, and model TDEP and STND are shown in Fig. 10. For almost all regions, the broad patterns are similar to those seen in dissolved $O_2$ concentration, with benthic and deep-water absolute $\delta^{13}C$ values generally similar to data for both model configurations. One exception is the Antarctic zone where $\delta^{13}C$ shows a good fit to data indicators nearer the surface, where modelled oxygen shows a poorer fit to data nearer the surface. The offset in mid-depth waters (~800m) in the sub-Antarctic zone may be due to a reduced Antarctic intermediate waters contribution in the model. This may also explain similar model-offsets at this depth in the South and East-Tropical Pacific regions. In warm surface waters, $\delta^{13}C$ reduces more quickly with depth in TDEP than STND, as nutrient recycling is occurring faster.

## 4.4 POC export and implications for biological carbon pump efficiency

The inclusion of a temperature-dependence term in remineralisation strongly affects both the export of POM via changes in the rate of nutrient recycling, as well as the efficiency of the biological carbon pump. To demonstrate the impact of each varied parameter, the export flux of POC (modelled at 80m) for every simulation (not only the best-fit TDEP) is shown in Fig. 11. With a lower activation energy requirement (low $E_a(1)$ value), less energy is needed for the remineralisation process to occur, this means nutrients are returned to surface waters more quickly, production is higher, and so POC flux at 80m is higher. Conversely, the higher the $E_a(1)$ value, the more energy is required to remineralise organic carbon. So, at higher $E_a(1)$, proportionally more organic carbon reaches depth making surface processes less important. The fraction of the POC exported that is recalcitrant and the maximum nutrient uptake rate at the surface becomes less important as $E_a(1)$ increases. This trend occurs because there is no variation in ocean temperature between runs in Figure 11 a), i.e., atmospheric $CO_2$ and climate are fixed. It is important to note however that a larger $E_a(1)$ leads to a larger sensitivity of remineralisation rates to changes in temperature, e.g., a higher $Q_{10}$ (Fig. 11 b) and c)). The $Q_{10}$ for remineralisation rates in Figure 11 ranges from below 2.3 ($E_a(1) = 53$ kJ mol$^{-1}$) to over 2.5 ($E_a(1) = 60$ kJ mol$^{-1}$) for a change in temperature from 0°C to 10°C (Fig. 11 c).

The remineralisation curves for each ocean region are shown in Fig. 12 for the best fit TDEP and STND model for POC (in gC m$^{-2}$ yr$^{-1}$). TDEP and STND have differing initial POC export fluxes with lower latitude warmer waters showing higher export in TDEP due to the increased nutrient recycling there. A dataset of POC flux (Mouw et al. 2016a) is overlaid on the remineralisation curves (Fig. 12). In both model configurations, the measured Antarctic zone POC flux at shallow and intermediate depths (< 1500m) is significantly lower than in the model. We do not account for iron limitation in the Southern Ocean (or elsewhere) in this particular configuration of the cGENIE model, which would tend to act to limit productivity and POC export there and hence could potentially explain some of the mismatch we observe at shallower depths. In general, the measured flux at depth appears reasonably represented with the exception of warmer regions, where the measured POC flux (e.g. east tropical Pacific, North Indian, east Tropical Atlantic) is generally higher than in the model. This likely reflects additional processes that may increase POC fluxes to depth such as ballasting by minerals (Klaas and Archer, 2002; Wilson et al 2012) and the lower reactivity of POC associated with increased recycling in low latitude plankton ecosystems (Aumont et al., 2017).

Overall, the pattern of the efficiency of the transfer of particles from 80m to 1040m (Fig. 13) in TDEP is similar to that found in Weber et al. (2016), where efficiency of transfer is essentially a measure of the rate of remineralistion; what fraction of the POC exported at 80m that reaches 1040m. Colder waters show higher transfer efficiency, with the lowest transfer efficiency seen in the sub-tropical gyres. The STND model has a fixed decay rate for all locations, so the transfer efficiency at any particular depth has a global uniform value. The global export-weighted mean remineralisation depth for STND is 627m, and for TDEP 378m ± 236m.

It should be noted that here we have included all available data from Mouw et al. (2016a) without any attempt to ensure these data are representative of the annual mean (where the model output represents the annual mean). POC flux measurements can be highly dependent on time of year and number of data measurement points available. Some of the model-data mismatch may then

be due to a mis-match between the interval in time represented by the data, and the annual mean of the model. For instance, blooms, which are not well represented in the model, may explain some of the very high POC flux values (for example 0.2 gC m$^{-2}$ yr$^{-1}$) in the North Atlantic and hence why the model annual mean appears to underestimate the flux.

## 5 Implications of including temperature-dependent microbial processes

### 5.1 The role of temperature in the marine carbon cycle response to historical warming

In this study we have focussed on the two main components of the biological carbon pump. Firstly, nutrient uptake rates due to the metabolic temperature-dependence of photosynthesising marine biota; secondly remineralisation rates of sinking particulate organic matter due to the metabolic temperature-dependence of respiring marine biota feeding on that sinking organic matter. We find that the calibrated temperature response of the respiration-based mechanism of remineralisation in the water column is more sensitive to temperature change (a mean $Q10$ of 2.28 over a range of temperatures from 0°C to 26°C, from Eq.5 using 54 kJ mol$^{-1}$ for $E_a(1)$) than the photosynthesis-based one (the Eppley curve has a $Q10$ of 1.88, in Eq. 3 and 4, Bissinger et al. 2008), in agreement with fundamental studies (Brown 2004). Historical temperature rise, which we induced in the cGENIE.muffin Earth system model by prescribing the observed $CO_2$ in the atmosphere, provides an illustrative example of the role and importance of including sufficient temperature-dependent processes in models. In this section we therefore discuss in more detail the transient differences between STND and TDEP model configurations.

Between the years 1700 and 2010, global mean air temperature in cGENIE increases by 0.94°C. In turn, warming at the ocean surface induces stratification in the water column, reducing nutrient re-supply to the surface from subsurface waters. In the STND model, this results in a pronounced drop in POC export at 80m of 2.9% (Fig. 14), in agreement with the average of the CMIP5 models (Bopp et al., 2013). However, the transfer efficiency is additionally affected in TDEP, with a drop of over 5% in the proportion of POC exported at 80m that reaches 1040m (equivalent to a shoaling of the global mean remineralisation depth of 16m) (Fig 15). This reduction in biological pump transfer efficiency is a result of increased rates of remineralisation in the warming water column, principally in surface and near-surface waters (while whole, volumetrically-weighted, ocean warming is 0.12°C over this period, 0.6°C occurs on a global mean basis in surface waters, and 0.02°C in deepest waters). The result is that for TDEP this stratification-induced nutrient-limitation effect on export is largely offset by the intensified recycling of nutrients in warmer surface and near-surface waters.

Between simulated pre-industrial and present-day model states, we found a substantially smaller drop in POC flux at 80m when temperature dependence was included (TDEP) compared to the standard model (STND). Global POC flux at 80m reduces by 0.3% between pre-industrial and present-day in TDEP, but with increases in the Southern Ocean of around 10% and in the tropics of around 1%, suggesting an increase in NPP (Net Primary Productivity) in the tropics. Kwiatkowski et al. (2017) identified a reduction in NPP with warming in the tropical ocean of 3±1% per degree of warming, based on responses to ENSO (El Nino Southern Oscillation) which on face value is inconsistent with our simulation of a possible increase in NPP in the tropics. Their

estimate utilised satellite based NPP products from data on chlorophyll and light incidence, and they found that in no data-constraint did NPP increase in the tropics (although the data constraint varied according to the NPP product used). However, Behrenfeld et al. (2015) noted that a reduction in chlorophyll does not necessarily represent a reduction in productivity, due to photoacclimation. The satellite based NPP products do not account or correct for this effect, so may well underestimate NPP in warming conditions. In an earlier study, Taucher and Oschlies (2011) found an increased NPP when temperature dependence was included in modelled future projections. Laufkotter et al. (2017) also found that when including a temperature-dependence and oxygen content-dependent remineralisation, NPP increased on warming due to intensified nutrient recycling in near-surface water. However, they suggested this was largely due to an initial positive bias in surface ocean nutrients. In a second model set-up, they reduced nutrient recycling in surface waters and find little impact on NPP between the temperature sensitive and temperature independent model in a future projection to 2100 CE.

In this study we make no distinction between dissolved oxygen and sulphate in terms of controlling the remineralisation rate of POC (unlike, for instance, Laufkotter et al. 2017). Cavan et al (2017) concluded that the large oxygen minimum zone in the Eastern Tropical Pacific reduces the rate of remineralisation due to the almost complete absence of zooplankton particle disaggregation within, and provides a negative feedback to warming. However, Cram et al. (2018) explained most of the regional variability in the flux of POC in the deep sea via particle size and the effect of temperature on remineralisation, with oxygen concentration providing a small improvement (by reducing nutrient recycling in the Eastern Tropical Pacific). Particle size plays a role in sinking speeds, as larger particles sink faster (generally), and particle size is a factor in export and transfer efficiency (Mouw et al 2016b). The configuration of cGENIE we employ here does not account for particle size and has a fixed sinking speed globally (by default, 125 m d$^{-1}$). The lack of particle size variability and oxygen concentration's role in remineralisation may explain some of the increased POC flux at 80m that the model shows since the pre-industrial period in tropical waters. This tropical POC export increase may also be partly due to initial higher nutrient concentrations compared to data, or to the increased remineralisation rates re-supplying nutrients to the surface, but may also be linked to changes in DOM cycling (see section 5.2). Predicted changes (and changes that may have already occurred) in NPP in low-nutrient warm waters are still subject to large uncertainties (Turner et al. 2015, Cross et al. 2015).

There is also still uncertainty as to the causes, and even patterns, in POC flux differences across different ocean regions (Henson et al., 2012; Marsay et al., 2015; Weber et al., 2016; Cram et al., 2018). We find the patterns of transfer efficiency (Fig. 13) for TDEP are in broad agreement with Marsay et al. (2015) and Weber et al. (2016). This transfer efficiency is not dependent on surface waters NPP patterns or on how much POC is exported at 80m in cGENIE, however, the absolute amount of carbon reaching the deep ocean does depend on NPP and export. On warming since the pre-industrial period we found a reduced POC flux at 80m as well as a reduction in transfer efficiency, combining to produce a reduction in the strength of the biological carbon pump with warming. This further implies an increased carbon pump strength in cooler climates, as per Heinze et al. (2016).

We note that circulation states and upwelling/downwelling changes can also have an impact on the distribution of carbon, oxygen and nutrients between the surface and the deep (Kvale et al 2019, Loptien and Dietze 2019), and are also model-dependent. Circulation changes are small between the pre-industrial and the present-day compared to the simulations in Kvale et al (2019)

where very high $CO_2$ (up to 1200ppm) and high surface temperature results in large ocean circulation pattern changes; increased nutrient storage in the deep ocean was due to longer residence time of deep ocean water in that study (see Chikamoto et al. 2008 for the effect of Atlantic Overturning Circulation shutdown in cGENIE). In our study we have found that the temperature-dependent biological pump offsets some of the effects of physical ocean response to warming (in increased near-surface nutrient recycling, so

offsetting the effect of increased ocean stratification that otherwise reduces surface nutrients in the STND simulation). However, this is not to suggest that a temperature-dependent biological pump could offset the effect of extreme changes in circulation, such as an AMOC shutdown, or for far more extreme warming scenarios than that applied here. We do not test such scenarios here.

## 5.2 Temperature and the cycle of dissolved organic matter in the ocean

In this paper and associated model calibration, we have focussed on the role of temperature in the production and fate of POM.

However, the relative partitioning of primary production into DOM (rather than POM) together with its mean residence time at (or close to) the ocean surface, modulates nutrient recycling. Temperature-sensitivity of these 2 processes thereby adds an additional set of feedbacks between climate and the biological carbon pump.

To explore the wider role of temperature in the marine carbon cycle and as a further option in the model, we add a similar temperature-dependence to the remineralisation of DOM as for POM (Equation 5):

$$k(T)_{DOM} = \beta_{DOM} e^{\left(-E_a / RT\right)} \qquad (7)$$

where $E_a$ is the activation energy and is assigned a value of 54000 J mol$^{-1}$, and $R$ and $T$ are the gas constant (J K$^{-1}$ mol$^{-1}$) and absolute temperature (K), respectively. $k(T)_{DOM}$ is the rate constant (year$^{-1}$) controlling the decay of DOM and replaces the previous fixed lifetime value of 2.0 year$^{-1}$.

The assumption that the same activation energy applies to DOM as for POM (i.e. setting $\beta_{DOM} = 9.0$ x $10^{11}$ in Eq. 7) leads

to a global mean DOM lifetime of approximately 0.02 years, compared to the value of 0.5 years in the standard model (Ridgwell et al., 2007) that follows Naijar et al. (2007). This might: (i) reflect a different mean quality (and hence a different activation energy) of the organic matter assumed to constitute DOM as opposed to POM; and/or (ii) reflect the dispersed nature of DOM versus the more concentrated POM and/or differences in the associated bacterial biomass; and/or (iii) that the assumed sinking speed of 125 m d$^{-1}$ is simply too unrealistically fast. However, to simplify the model re-tuning of the DOM cycle, we only adjust the value of

$\beta_{DOM}$. For a fixed production fraction of 0.66, setting $\beta_{DOM} = 1.16$ x $10^{10}$ gives a mean (flux weighted) DOM lifetime in the model that matches the fixed 0.5 year value of the original model configuration.

The production of DOM (vs. POM) is also likely to be influenced by ambient temperature. Dunne et al. (2005) describe a multiple linear regression analysis of observed ocean surface properties, as well as primary and export production, across a range of different ocean environments, and deduce a role for sea surface temperature in predicting the observed particle export ratio.

Although the Dunne et al. (2005) regression based on temperature and net primary productivity has previously been employed by Ma and Tian (2014) to formulate the production of DOM vs. POM (and co-incidentally, also in a version of the cGENIE Earth system model), the default biological scheme of 'induced fluxes' (Maier-Reimer, 1993) does not provide a value of primary

productivity – required by the regression model. Additionally, the default configuration of the ocean circulation model does not calculate a mixed layer depth – required to convert between units for primary production in the regression formula. Dunne et al.
(2005) also provide an alternative and slightly improved regression model based on mixed layer Chl$a$ concentrations (and temperature), but this would require use of the Ward et al. (2018) 'ECOGEM' ecological model component in the cGENIE Earth system model, and outside the scope of this present study.

Rather than attempt to reformulate (and re-fit) the Dunne et al. (2005) regression model as a function of the environmental parameters simulated by the cGENIE model, we simply extract the temperature sensitivity term in isolation (while recognising that
is was derived by Dunne et al. (2005) jointly alongside primary production (or Chl$a$)) in order to derive a temperature-dependent equation for the partitioning of POM vs. total organic matter export (DOM+POM), $\gamma$:

$$\gamma = c - dT \qquad\qquad for\ 0.04\ <\gamma<\ 0.72 \qquad\qquad (8)$$

where $d$ is the temperature sensitivity of 0.0101 °C$^{-1}$ from Dunne et al. (2005), and $c$ is a constant (0.512), whose value is chosen such that the global mean (production-weighted) export of DOM vs. POM occurs in a 2:1 ratio (i.e. $\gamma = 0.34$).

This completes the implementation of an option for temperature-dependency in both the creation and remineralisation of DOM, with the exception that the two parameterisations interact, and in order to configure cGENIE such that the mean production fraction of DOM and mean lifetime both align with the values (0.66 and 0.5 years, respectively) in the original model (i.e. Ridgwell
et al., 2007; Cao et al., 2009), we make a final adjustment to: $\beta_{DOM} = 1.32 \times 10^{10}$ (and keeping the initial calibrated value of $a = 0.512$ in Eq. 8).

A further simulation (a pre-industrial spin-up followed by a transient simulation forced by $CO_2$ and $\delta^{13}CO_2$ data-indicated changes from 1700 to the year 2010) - called TDEP$_{+TDOM}$ - was conducted that includes the best fit TDEP parameters but now adds the temperature-dependent production and decay of DOM as described above. For the present day, TDEP$_{+TDOM}$ slightly improves
the model fit to oxygen distribution compared to TDEP, with lower CRMSD and higher correlation, whilst model fit to PO$_4$ is slightly worsened (Table 4).

 The inclusion of temperature-dependent DOM processes also affects the response of ocean carbon cycling to historical warming. The global mean export-weighted remineralisation depth for the present-day in TDEP$_{+TDOM}$ is 399m +- 255m (21m deeper than TDEP), and the change in depth since 1700 is a shallowing of 16m (the same as for TDEP). Slightly less POC is exported at
80m in TDEP$_{+TDOM}$ compared to TDEP by 2010 (Fig.14), indicating that although increases in surface nutrient recycling (due to temperature-dependent POC processes) significantly offsets the effects of warming-induced ocean stratification in TDEP, this is counterbalanced but to a lesser extent by temperature dependence in DOM production and remineralisation in TDEP$_{+TDOM}$.

In response to warming and compared to STND and TDEP, TDEP$_{+TDOM}$ exhibits a slightly increased global DOM production ratio (the fraction of organic matter produced that is dissolved rather than particulate, as Fig 1) driven by temperature-
dependent DOM production (table 5). This is countered, however, with a (global) shorter DOM lifetime at the surface driven by temperature-dependent DOM remineralisation. The net result is a decline of 0.74% in the mean global surface DOM concentration

between 1700 and 2010 CE in TDEP$_{+TDOM}$. In comparison, there is an increase of 0.54% in mean global surface DOM concentration for TDEP and a decrease of 2.07% in STND (Tables 5 and 6).

Appendix figures A1 and A2 provide a view of the spatial changes in DOM dynamics taking place between 1700 and 2100, and for each DOM process being individually temperature-enabled as well as both together. A summary is that the effect of temperature-dependent DOM is to increase the DOM production ratio in low latitudes, and decrease the production ratio in high latitudes compared to TDEP, with historical warming resulting in a general increase in DOM production between 1700 and 2010 in TDEP$_{+TDOM}$. The TDEP$_{+TDOM}$ DOM lifetime at the surface is decreased in low latitudes and increased in high latitudes compared to TDEP, with a general decrease in DOM lifetime in response to historical warming, and more pronounced in higher latitudes. TDEP$_{+TDOM}$ has lower surface DOM concentrations in the tropics compared to TDEP in the present day. Since 1700, surface DOM concentration has increased in the southern high latitudes in all model configurations, but TDEP$_{+TDOM}$ results in a slight decreased surface DOM concentration in the tropics, compared to a slight increase in TDEP.

Overall, the combined effects of temperature-dependent POM and DOM processes still results in a smaller reduction in modelled POC export at 80m compared to STND (Fig. 14 a) of 0.7% since 1700, although when only POM processes are enable as temperature-dependent, the decrease is smaller still (0.3%). Compared to TDEP: the TDEP$_{+TDOM}$ shows lower export production in mid and high latitudes (Fig. 14 b) but affects almost no change in transfer efficiency of the biological pump (Fig 15).

## 6 Summary

The STND and TDEP variants of the cGENIE.muffin model are tuned to observed PO$_4$ and O$_2$ distributions and achieve comparable statistical fits to ocean climatologies. However, substituting temperature-dependent particulate organic matter (POM) export and remineralisation parameterisations into cGENIE.muffin changes the patterns of PO$_4$, dissolved oxygen, and carbon-13 distributions in the ocean compared to the standard model, and substantive differences between models occur in response to warming since the pre-industrial period. Specifically, in response to warming, inclusion of a temperature-dependency on the production and remineralisation of POM increases the efficiency of nutrient and carbon recycling to the surface. In the case of a transient and geologically-rapid warming, this temperature-driven increase in recycling is sufficient to largely offset a decrease in nutrient re-supply due to enhanced physical stratification of the upper ocean, leaving global POM export relatively unaffected by historical warming. Additionally accounting for temperature-dependency in both the production and remineralisation of dissolved organic matter (DOM) further modifies the global carbon cycle response to historical warming, but to a much lesser extent than did the inclusion of POM-linked processes. Finally, the temperature-dependent biological pump results in a large reduction in the efficiency of the transfer of carbon from the surface to the deep ocean over the historical period simulation compared to the standard model.

**Appendix A, Further information on the temperature-dependent DOM**

We combined the processes of temperature-dependent uptake and remineralisation for the POC driven biological pump as each process has been individually applied in previous published work. However, for the newly developed temperature-dependent DOM processes we show here (for each model configuration discussed in the text) the separate effects of temperature-dependent DOM production and temperature-dependent DOM remineralisation mapped as global surface distributions for: DOM production ratio, DOM lifetime, and DOM surface concentration (Fig A1). We further show for each parameter the anomaly at 2010 with respect to the year 1700 (Fig A2).

**7 Model code availability**

The specific version used of the cGENIE.muffin model used in this paper is tagged as release v0.9.13, and has been assigned a DOI: 10.5281/zenodo.3999080. The code is hosted on GitHub and can be obtained by cloning:

https://github.com/derpycode/cgenie.muffin

changing the directory to cgenie.muffin and then checking out the specific release:

$ git checkout v0.9.13

Configuration files for the specific experiments presented in the paper can be found in the directory:

genie-userconfigs\MS\crichtonetal.GMD.2020

Details of the experiments, plus the command line needed to run each one, are given in the readme.txt file in that directory. All other configuration files and boundary conditions are provided as part of the release.

A manual, detailing code installation, basic model configuration, plus an extensive series of tutorials covering various aspects of muffin capability, experimental design, and results output and processing, is provided on GitHub. The latex source of the manual, along with pre-built PDF file can be obtained, by cloning:

https://github.com/derpycode/muffindoc

**Author contribution**

KAC devised, ran and analysed the model ensemble and data, JDW and AR developed the temperature-dependent remineralisation process, AR developed the temperature-dependent DOM processes, all authors wrote the manuscript.

**Acknowledgments**

KAC was supported by Natural Environment Research Council (NERC) grant number NE/N001621/1 to PNP. JDW was supported by the European Research Council as part of the "PALEOGENiE" project (ERC-2013-CoG-617313). AR acknowledges additional
support from the National Science Foundation under Grants 1702913 and 1736771.

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

|  | Rate constant as T approaches infinity (*a*) | Multiplier constant for T (*1/b*) | Reference |
|---|---|---|---|
| Eppley curve | 0.59 | 0.0633 | Eppley et al. 1972 |
| LPD curve | 0.81 | 0.0631 | Bissinger et al. 2008 |

**Table 1, Values for variables in Eq. 4**

| Name | Circulation | Biochemistry | 1. Temperature-dependent uptake | 2. Temperature-dependent remineralisation | Temperature-dependent DOM (both production and | Description |
|---|---|---|---|---|---|---|
| STND | ✓ | ✓ |  |  |  | Standard model |
| TDEP | ✓ | ✓ | ✓ | ✓ |  | Temperature-dependent POM model |
| TDEP$_{+TDOM}$ | ✓ | ✓ | ✓ | ✓ | ✓ | Temperature-dependent POM and DOM model |

**Table 2, Model settings, processes included in each set-up. Column numbering corresponds to numbering in figure 1.**

| Parameter | Values applied |
|---|---|
| Vmax | 4,7,10 |
| POC fraction 2 (recalcitrant) | 0.002, **0.008**, 0.032 |
| Ea(1) (labile fraction) x10$^3$J/mol | 53, 54, **55**, 56, 60 |

**Table 3, settings for parameters in TDEP (temperature-dependent). Values in bold are those applied in John et al. 2014.**


| | Standard Deviation | | | | CRMSD | | | correlation | | |
|---|---|---|---|---|---|---|---|---|---|---|
| | Data | STND | TDEP | TDEP+TDOM | STND | TDEP | TDEP+TDOM | STND | TDEP | TDEP+TDOM |
| Whole ocean PO$_4$ | 0.8072 | 0.8072 | 0.7386 | 0.7606 | 0.2208 | 0.2043 | 0.2113 | 0.9236 | 0.9227 | 0.9177 |
| Surface PO$_4$ | 0.6026 | 0.5009 | 0.4452 | 0.3921 | 0.1699 | 0.1820 | 0.2130 | 0.9350 | 0.9294 | 0.9274 |
| Whole ocean O$_2$ | 1.6461 | 1.5154 | 1.3900 | 1.4522 | 0.5476 | 0.5501 | 0.5356 | 0.8670 | 0.8615 | 0.8703 |
| O$_2$ 283m to 411m | 1.7285 | 1.3724 | 1.6531 | 1.6142 | 0.8145 | 0.8443 | 0.8200 | 0.7551 | 0.7660 | 0.7746 |

**Table 4, Statistics for the fit of the model to phosphate and oxygen distribution in the present day for the STND, TDEP (best-fit) and TDEP+TDOM simulations. Statistics shown as standard deviation (SD) in µmol/l, centred root mean square difference (CRMSD) in µmol/l, and correlation to data. TDEP+TDOM is described in section 5.2.**

| | DOM production ratio | surface DOM lifetime (yr) | Surface DOM concentration (mol/kg) |
|---|---|---|---|
| STND | 0.6600 | 0.5000 | 11.5 x 10$^{-6}$ |
| TDEP | 0.6600 | 0.5000 | 17.7 x 10$^{-6}$ |
| TDEP +DOMtprod | 0.6685 | 0.5000 | 20.3 x 10$^{-6}$ |
| TDEP +DOMtremin | 0.6600 | 0.4753 | 13.9 x 10$^{-6}$ |
| TDEP+TDOM | 0.6677 | 0.4907 | 13.4 x 10$^{-6}$ |

**Table 5, Present (2010) DOM parameters.**

| | DOM production ratio change (%) | surface DOM lifetime change (%) | Surface DOM concentration change (%) |
|---|---|---|---|
| STND | 0.00 | 0.00 | -2.07 |
| TDEP | 0.00 | 0.00 | 0.54 |
| TDEP+DOMtprod | 0.99 | 0.00 | 2.89 |
| TDEP+DOMtremin | 0.00 | -4.38 | -2.52 |
| TDEP+TDOM | -0.33 | -0.75 | -0.74 |

**Table 6, Anomaly (as %) DOM parameters for 2010 with respect to 1700.**

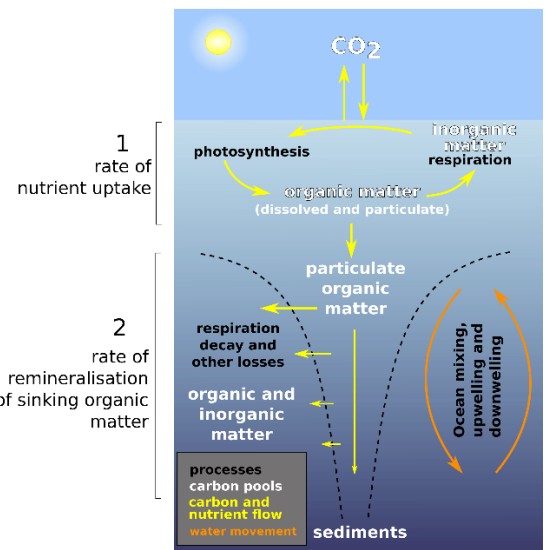

**Figure 1, Simplified schematic of the ocean biological pump and dissolved nutrient movements, and the two temperature-dependent processes that are considered in this study 1. Nutrient uptake rate, 2. Remineralisation rate. In the style of U.S. DOE (2008). We do not model sediments in this study, but it appears in the figure for completeness.**

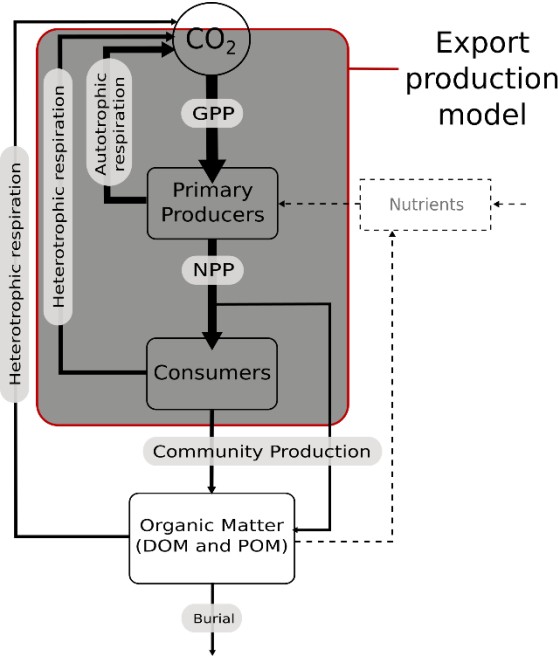


**Figure 2. Schematic of biological pump processes showing where cGENIEs export production operates. In the export production model, no mechanistic consideration of the effects of temperature within the mixed-layer (i.e. GPP vs NPP vs community production) can be considered, but heterotrophic respiration (as remineralisation) vs community production (as export production) can be considered, as well as nutrient recycling. In this study we apply temperature dependency to Organic Matter production and remineralisation that drives**
**the biological carbon pump. We do not model burial in this version of cGENIE (but it is here for completeness). In this cGENIE configuration, the nutrient is phosphate. Dashed line indicates the cycling (and re-supply due to circulation) of nutrients. Solid lines indicates the cycling of carbon.**

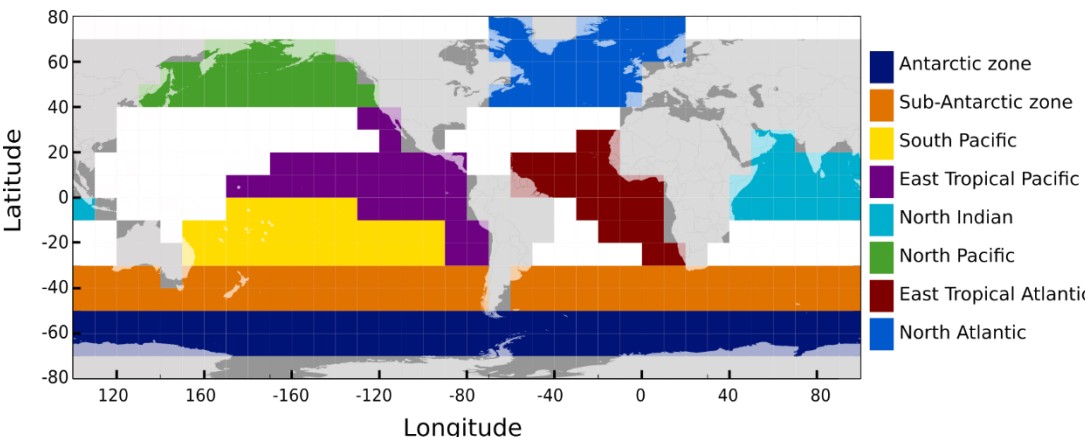

**Figure 3, Selected ocean regions for model-data comparison (on a 10x10 degree grid, with land masses overlaid for indication), based on Weber et al. (2016) and WOA 2009 data (Levitus et al., 2010).**

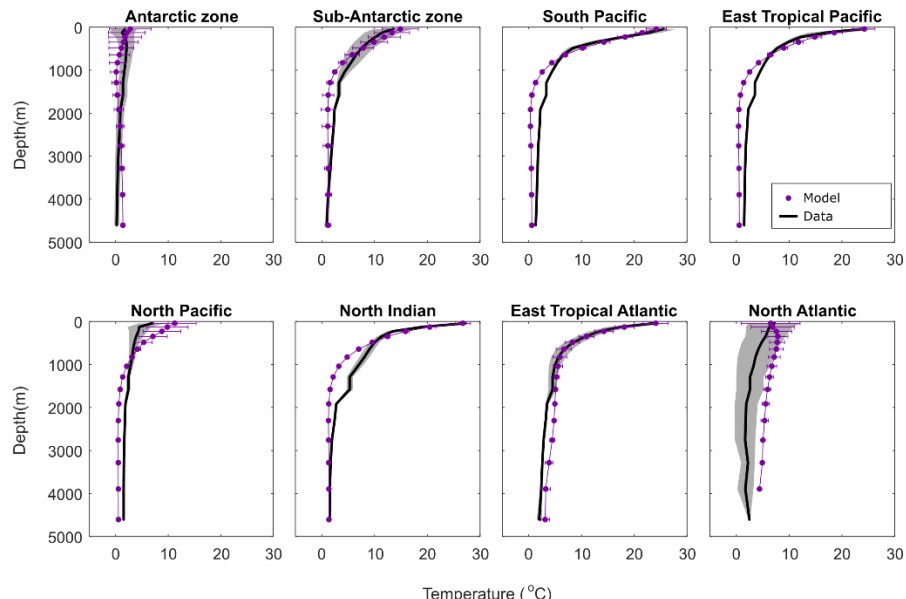

**Figure 4, Temperature (°C) per depth by region for model and data (mean and standard deviation). Data from WOA 2009 (Levitus et al., 770 2010).**

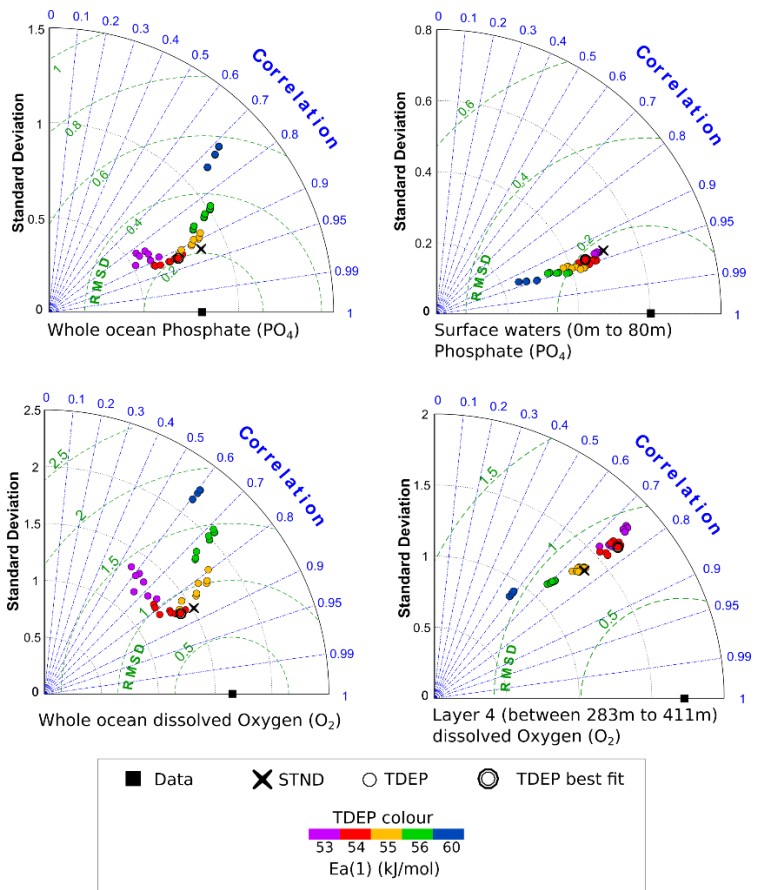

**Figure 5, Taylor diagrams for model fit to data for PO₄ and O₂ concentrations, showing standard deviation (standard deviation is not normalised), correlation and centred root means squared difference (RMSD). Data from WOA 2009 (Levitus et al., 2010). The best fit TDEP setting is double circled.**

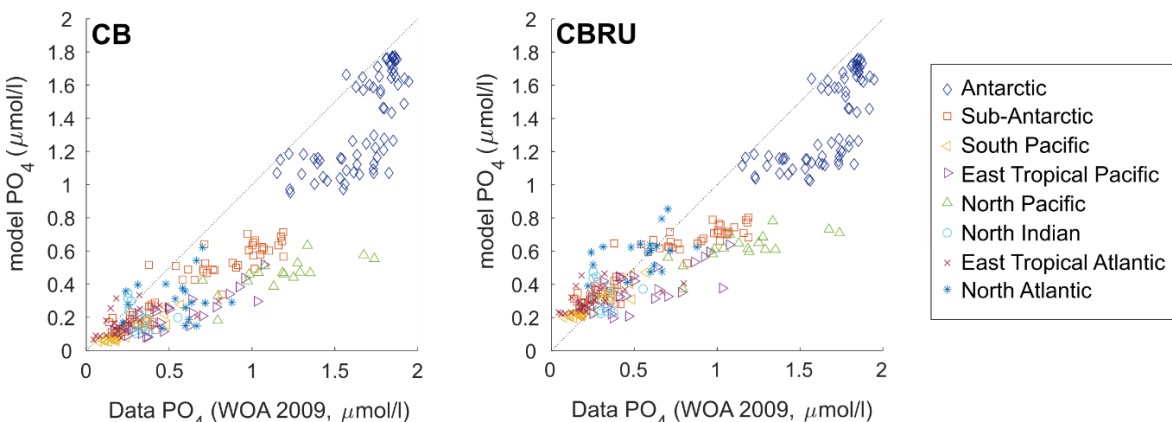

**Figure 6, Cross plot for surface (0m to 80m) PO₄ concentrations (µmol l⁻¹) for data and model labelled by ocean region. Data from WOA 2009 (Levitus et al., 2010)**

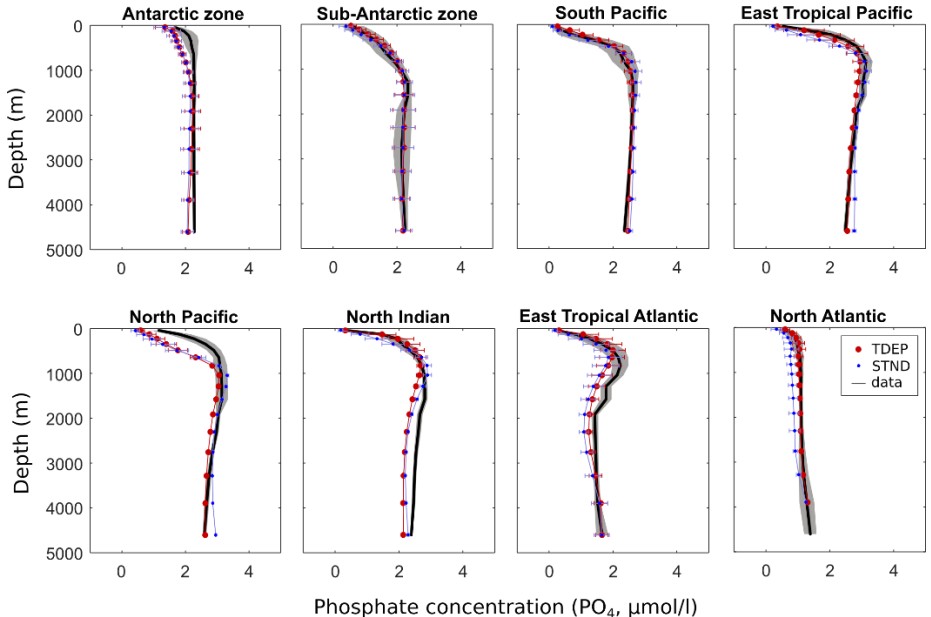


**Figure 7, PO₄ (µmol l⁻¹) per depth by region for model and data (mean and standard deviation). Data from WOA 2009 (Levitus et al., 2010). STND is standard model, TDEP is temperature-dependent model.**

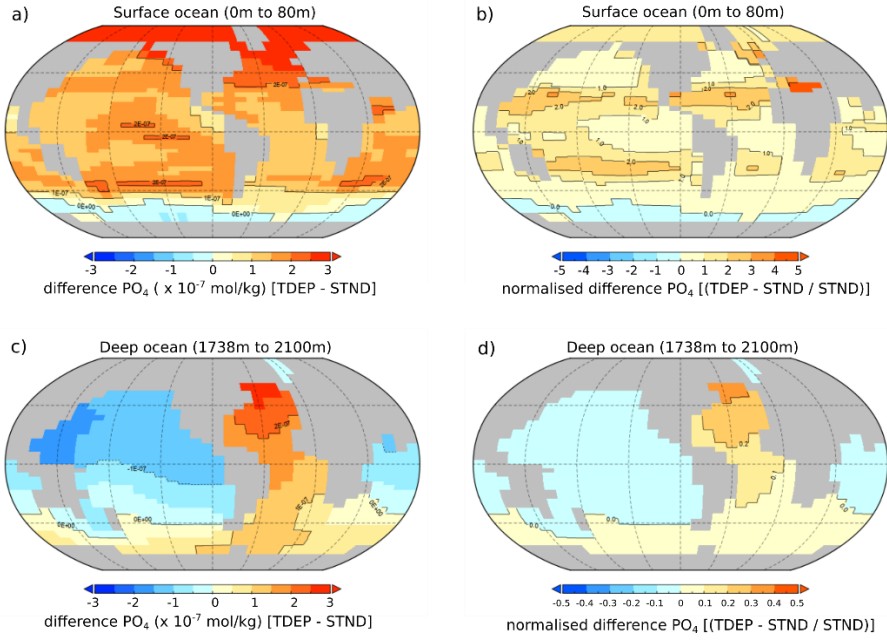


**Figure 8, Difference in PO₄ concentration in best-fit TDEP compared to STND, a) and b) surface waters (0m to 80m), c) and d) deep waters 1738m to 2100m). Left a) and c) absolute difference (mol/kg), right b) and d) normalised difference. All are the present-day, note scale difference on normalised difference between surface and deep. STND is standard model, TDEP is temperature-dependent model.**

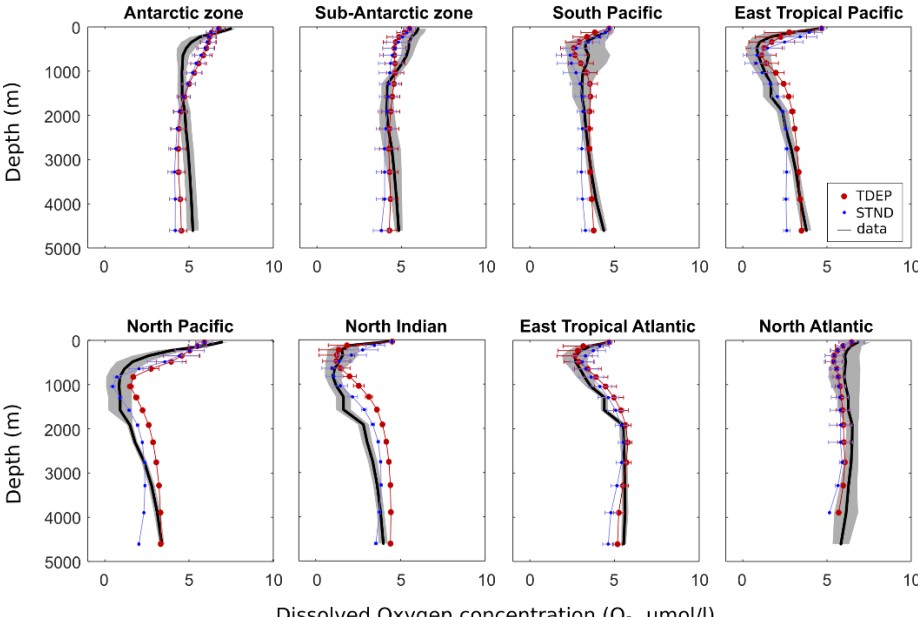

Figure 9, Dissolved $O_2$ ($\mu$mol l$^{-1}$) per depth by region for model and data (mean and standard deviation). Data from WOA 2009 (Levitus et al., 2010). STND is standard model, TDEP is temperature-dependent model.

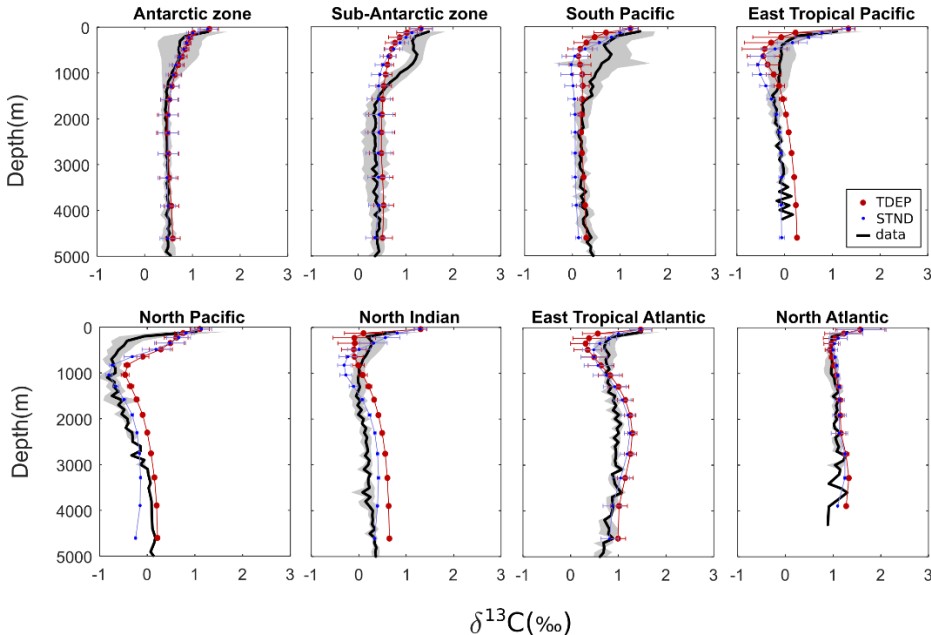

Figure 10, $\delta^{13}$C of DIC (‰ VPDB) per depth by region for model and data (mean and standard deviation). Data from Schmittner et al. (2013). STND is standard model, TDEP is temperature-dependent model.

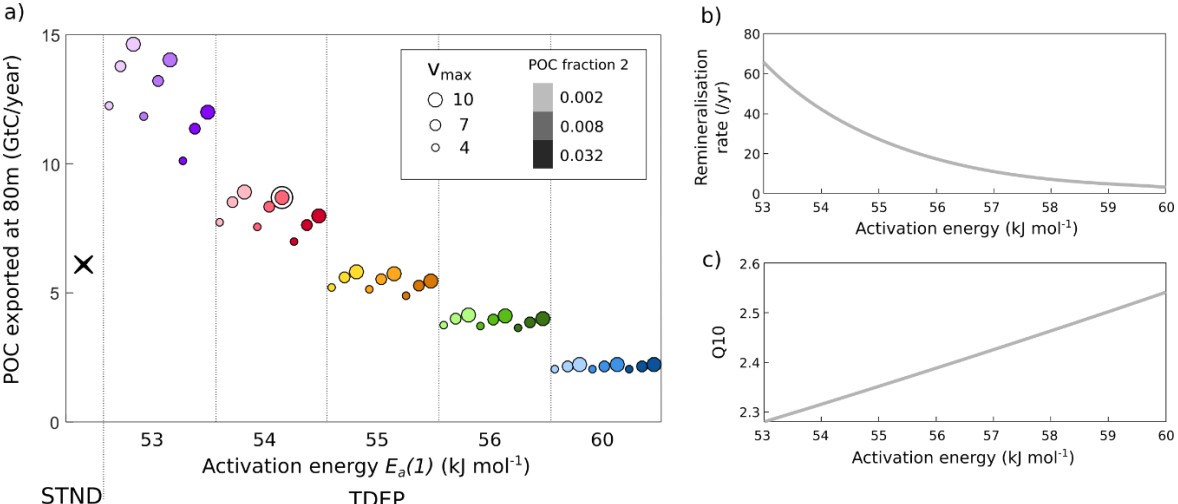

**Figure 11, Global POC flux (GtC yr⁻¹) at 80m (a) and the effect of activation energy on (b) remineralisation rate and (c) Q10 (for a 0°C to 10°C change). In a) STND model is shown as a black cross, TDEP model are circles. Best fit TDEP setting is double circled. STND is standard model, TDEP is temperature-dependent model. Colours in a) reflect those used in figure 5.**


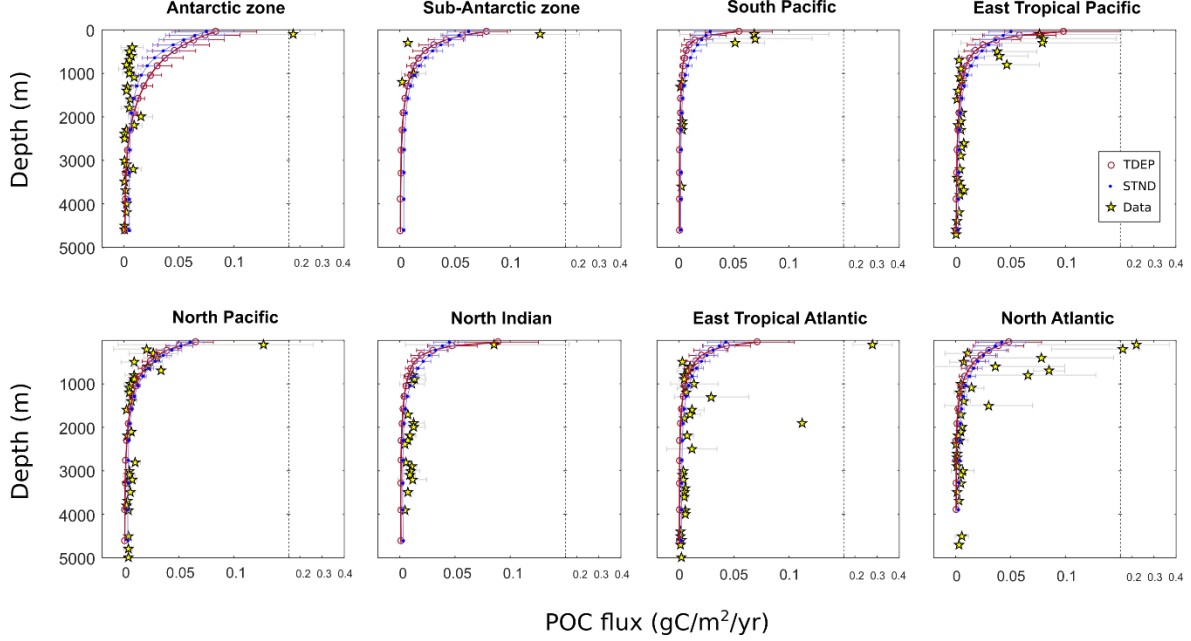

**Figure 12, POC flux (gC m⁻² yr⁻¹) for model (mean and standard deviation) and data. Data from Mouw et al. (2016a). STND is standard model, TDEP is temperature-dependent model.**

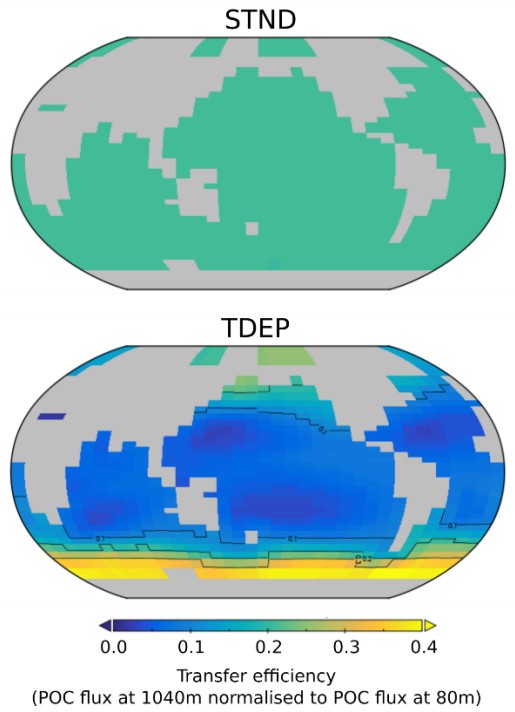

STND

TDEP

Transfer efficiency
(POC flux at 1040m normalised to POC flux at 80m)

**Figure 13, Model POC transfer efficiency (also used as a measure of biological carbon pump efficiency here) for STND (top) and best fit TDEP (bottom). Transfer efficiency is the fraction of POC exported at 80m that reaches 1040m, for the year 2010. Global transfer efficiency value for STND is 0.208.**

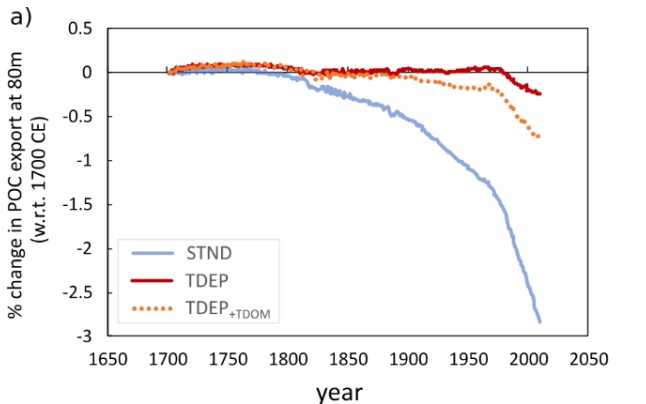

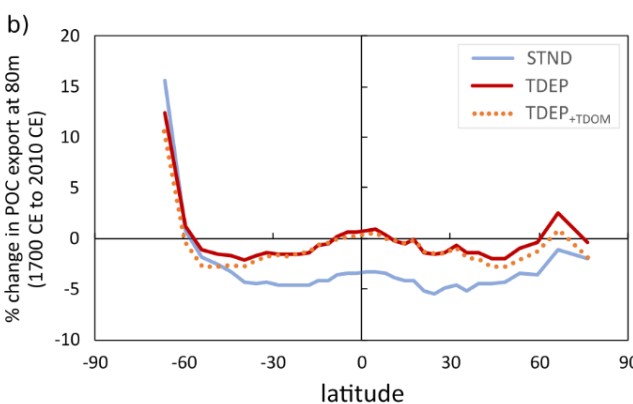

**Figure 14, POC export at 80m, % change with respect to (w.r.t) the year 1700, a) global mean POC export at 80m, b) latitudinal mean POC export change at 2100 w.r.t. 1700CE. STND is standard model, TDEP is temperature-dependent model, TDEP+TDOM is temperature-dependent POM and DOM and is described in section 5.2.**

a)

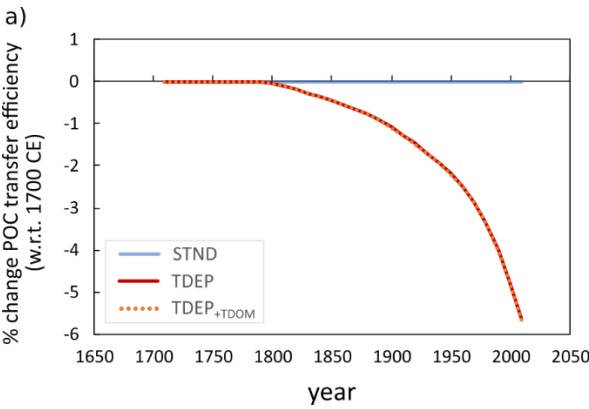

b)

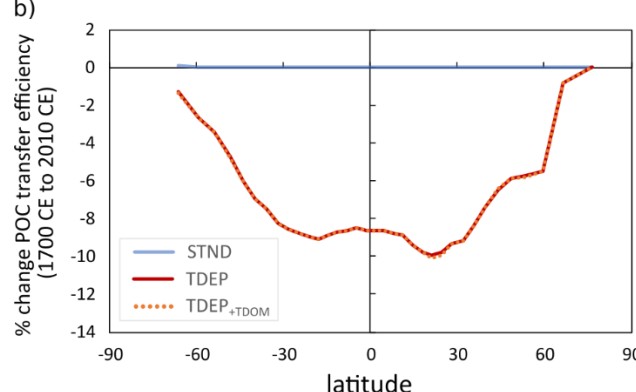


**Figure 15, TDEP biological pump "transfer efficiency" (the proportion of POC exported at 80m that reaches 1040m) % change with respect to 1700 CE., a) global mean change per year w.r.t 1700; b) latitudinal change at the year 2010 w.r.t 1700 . STND is standard model, TDEP is temperature-dependent model, TDEP+TDOM is temperature-dependent POM and DOM and is described in section 5.2.**


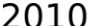

2010

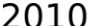

**Figure A1, Surface ocean DOM production ration, DOM lifetime and DOM concentration for the modelled present day (year 2010). Labelled simulations include those discussed in the main text (STND, TDEP and TDEP+TDOM), TDEP+TDOM labelled here as TDEPTDOM(PROD+REMIN) for clarity. Also mapped is DOM production and remineralisation temperature dependence separately applied as TDEP+TPROD DOM TDEP+TREMIN DOM).**

# Anomaly from 1700

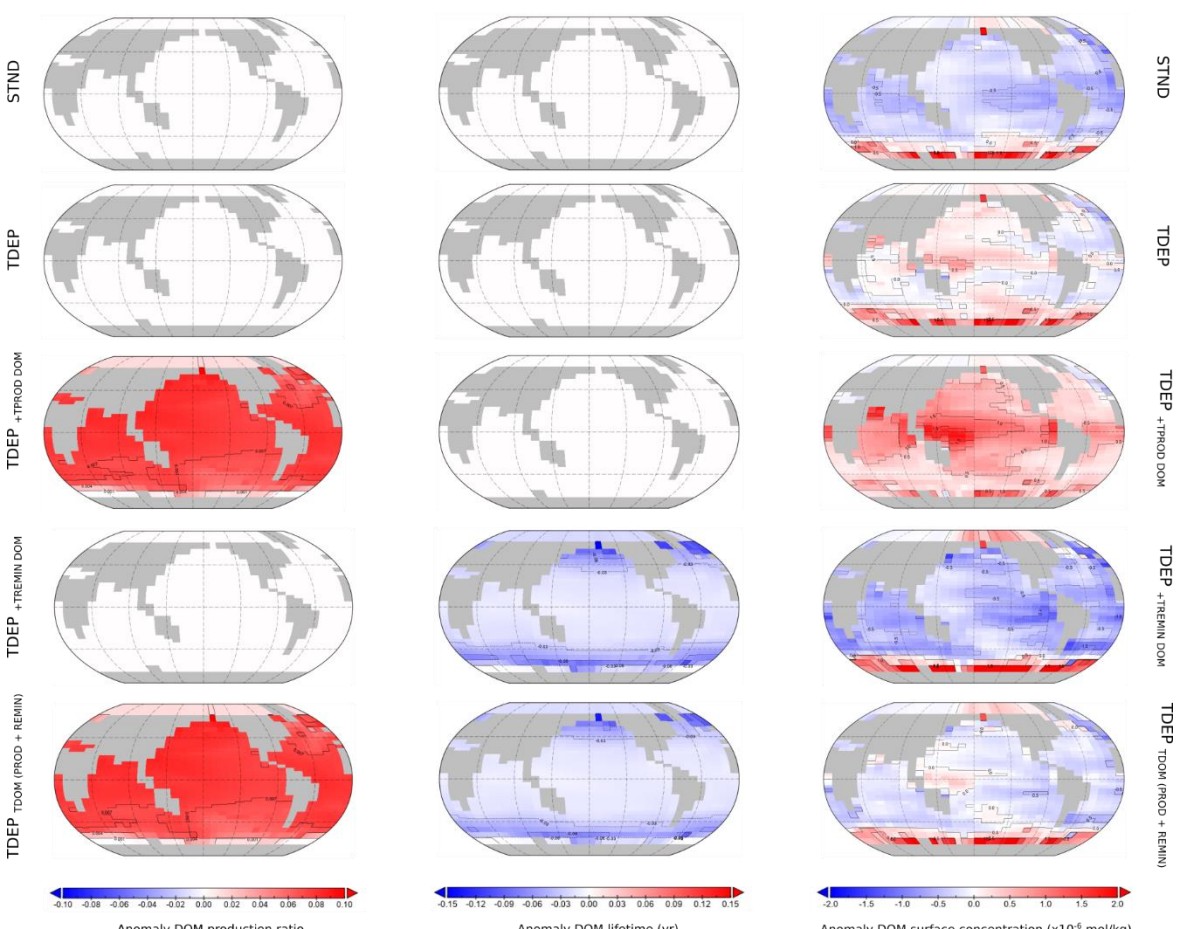

**Figure A2, Anomalies for the present day against the year 1700 for surface ocean DOM production ratio, DOM lifetime and DOM concentration for the modelled present day. Labelled simulations include those discussed in the main text (STND, TDEP and TDEP$_{+TDOM}$), TDEP$_{+TDOM}$ labelled here as TDEP$_{TDOM(PROD+REMIN)}$ for clarity. Also mapped is DOM production and remineralisation temperature dependence separately applied as TDEP$_{+TPROD\,DOM}$ TDEP$_{+TREMIN\,DOM}$).**