# Peer review of "Calibration of temperature-dependent ocean microbial processes in the cGENIE.muffin (v0.9.13) Earth system model"

_Geoscientific Model Development, 2019_

## Referee Comment (RC1) · Anonymous Referee #1 · 14 Mar 2020

General comments:

This study adds temperature dependency to the GENIE EMIC biological export and remineralization parameterizations, calibrates key parameters against observational data, and compares model biological pump characteristics and behavior to the previous temperature-independent version. Temperature dependency slightly improves the model against data but produces significant changes to transient behavior of the biological pump in response to warming. The application of temperature dependency for ocean biological processes in earth system models is generally recognized to be important, but updating the current suite of models already in use takes time and effort.

[Figure]

This paper quantifies the importance of temperature dependency over the historical time period since 1700 for one of the more popular EMICs and is therefore relevant for GMD. The study is not a substantial advance, but uses valid methods and proposes an elegant parameterization that could be applied to other models, as well has having potential interesting further applications in GENIE. The results support the interpretations, although some of the conclusions should be reconsidered (see below). The material is presented clearly and in a reproducible fashion and the model code has been made available.

Specific comments:

Figure 2 highlights a key opportunity that has been overlooked in this study, which is that the fractional assignment of export to the DOM pool could easily be made temperature-dependent, and that would implicitly represent temperature-dependent fast recycling processes in the upper ocean. The authors should discuss why this parameter was not included in their calibration.

Application of temperature dependency to export and remineralization will have also affected turnover in the DOM pool, but this is never discussed or shown in the manuscript.

There are some relevant references missing from the manuscript:

In the Introduction:

Schmittner et al. 2008, GBC doi:10.1029/2007GB002953 Introduced a temperature-dependent remineralization parameterization to the UVic ESCM. The Hülse paper referenced in the present manuscript was not correct in that there is temperature-dependency in the interior ocean biological pump in the UVic ESCM (see Schmittner et al. 2008, equation A16). Unless explicitly stated, every UVic ESCM ocean carbon study since Schmittner et al., 2008 has used temperature-dependent remineralization (the rate of remineralization is temperature-dependent, while the sinking rate of detritus is depth-dependent), as well as a temperature-dependent microbial loop and

temperature-dependent primary production and mortality, and since:

Keller et al. (2012) doi:10.5194/gmd-5-1195-2012 temperature-dependent zooplankton growth and grazing.

In Section 2, page 4, lines 103-110:

Kvale et al. 2015, ERL doi:10.1088/1748-9326/10/7/074009 This study explored the sensitivity to the NPP:respiration ratio and export efficiency response in a warming scenario to the application of mineral ballast in the UVic ESCM. The effect on export from ballasting inhibiting temperature-enhanced remineralization is arguably not secondary.

Kvale et al. 2019, BG /10.5194/bg-16-1019-2019 This study extended Kvale et al. 2015 to look at biological pump response to warming from an icehouse to a greenhouse state, and cooling from a greenhouse to an icehouse state. It demonstrates steady-state nutrient storage in warmer and cooler climates is counter to what is proposed in the Summary section for reasons of circulation.

In the Summary (or Introduction): Löptien and Dietze (2019) BG 10.5194/bg-16-1865-2019 This study demonstrates the increasing sensitivity of an earth system model to temperature-dependent biological processes and compensory model tuning with warming.

Technical corrections:

P2L49: "homogenizing the ocean interior"

P2L51: What does "mean" mean in this context? Annual mean? Global mean is a crude approximation, after all this study adds local temperature effects.

P3L75: "However, higher temperatures lower CO2 solubility. . ."

P3L78: "reduced"

P3L80: "temperature"

P3L91: "...but is not employed..."

P6, Equation 5: Please either rename A or make it clear on this page that the value is different than the A in Equation 4.

P7L89: Meyer et al. (2016) prescribed several e-folding depths, which approximates, but is not the same as, temperature dependence in export

P7L197: "dominates export"

P7L198: "nutrient" should be "PO4"

P7L201: please give units for Vmax

P8L246: "low latitude"

P9L250: the North Pacific subsurface temperature profile is over-estimated in GENIE according to Fig 4, please correct this sentence

P9L256: What are the lowest RMSE for CB and CBRU?

P9L261: "nutrient" should be "PO4"

P9L268: My understanding is the models are also tuned to O2 (from P9L253-256)

P9L272: "...the Southern Ocean..."

P10L285: "circulation difference" should be "circulation control" or "circulation dominance"

P12L345: "the observed CO2 transient" is awkward phrasing

P12L375: what is the Eastern Tropical North Pacific?

P13L380: "tropical POC"

P13L380-381: DOM cycling is also changed

P13L392: "nutrient" should be "PO4"

P13L393: remove "in some way"

P13 Summary: Please see Kvale et al. 2019 BG /10.5194/bg-16-1019-2019

P19, Table 2 can be cut at no detriment to the manuscript. 'CB' and 'CBRU' are defined too late in the manuscript and the naming includes an extra (confusing) reference to BIOGEM. I suggest omitting the Table and renaming the simulations to something more descriptive, like 'Temp' and 'NoTemp', since all models contain 'Remineralization' and 'Uptake'.

Table 3, Figs. 5 and 11: parens missing on Ea(1)

P20, Figure 1 can be cut at no detriment to the manuscript.

P21, Figure 2 caption: is "mixed player plankton" supposed to read "mixed phytoplankton"? Please add a key to clarify what dashed/solid/thick/thin lines, and shading, represent. Why is burial shown if there are no sediments? Should "nutrients" be PO4 (only PO4 in this model)? Why are autotrophic respiration/heterotrophic respiration/consumers shown if they are not included in the model?

P23, Figure 8. I see why the figure is normalized (the point made on P9L275), but normalization is misleading (small differences of low concentrations appear to be significant). The figure would be more informative presented without normalization, but the above point can still be made in the text.
* * *

---

## Short Comment (SC1) · 26 Mar 2020

Dear authors,

in my role as Executive editor of GMD, I would like to bring to your attention our Editorial version 1.2:

https://www.geosci-model-dev.net/12/2215/2019/

This highlights some requirements of papers published in GMD, which is also available on the GMD website in the 'Manuscript Types' section:

http://www.geoscientific-model-development.net/submission/manuscript_types.html

[Figure]

In particular, please note that for your paper, the following requirement has not been met in the Discussions paper:

- "The main paper must give the model name and version number (or other unique identifier) in the title."

Please add a version number (v0.9.7) for cGENIE.muffin to the title upon your revised submission to GMD. Yours,

Astrid Kerkweg

—————————————————————

---

## Referee Comment (RC2) · Anonymous Referee #2 · 3 Apr 2020

General comments This study presents the importance of modeling temperature-dependent microbial processes in the existing EMIC, evaluates the model performance using multiple tracer variables, and discusses the impact of the modified processes on marine carbon cycling under changing climates. The study employs a sophisticated approach for joint parameter tuning and shows that temperature-dependency introduces a significant difference (improvement) in biological pump efficiency estimates. Redeveloping a global biogeochemical model takes time and effort, and there is a limit to increasing complexity in the model due to computational- and data validation issues. Yet, this study presents significant findings by modifying the existing nutrient uptake and remineralization schemes without adding temperature-dependency into all different levels of biological processes and trophic interactions. Overall, the model experimental design is constructive, and the manuscript is well-written, easy to understand, and relevant for GMD.

Specific comments Section 4.4 (line 308-315) and Figure 11 provide mechanistic insight into why CBRU results are different from CB results via the effect of model parameter variation on the model output. Though automated data assimilation or parameter optimization was not conducted in the study, the best parameter set that consists of Vmax, rPOM, and Ea(1) was indirectly determined based on the model-observation correspondence of multiple tracer variables. Compared to Vmax and rPOM, Ea(1) drives a much larger variation in POC export ($\sim$7 kJ/mol increases of Ea(1) drives $\sim$13 GtC/year decreases of POC export). Under higher Ea(1) values the sensitivity of POC export to Vmax and rPOM decreases, and Ea(1) higher than 54 kJ/mol might not be the "right" parameter value. At Ea(1) of >54.5 kJ/mol POC export becomes lower compared to the non-temperature dependent model. This could be concerning given that a small variation of the Ea(1) value ($\sim$0.5 kJ/mol) results in completely opposite patterns compared to the findings of the study, i.e., the temperature-dependent model simulates lower POC export than the non-temperature dependent model. The uncertainty related to this finding does not come across clearly and should be discussed. Also, it does not seem necessary for circles to be color-coded to reflect different rPOM in Figure 11.

In Introduction, in relation to "A deeper mean remineralization depth equates to a more "efficient" biological carbon pump" it would be good to calculate the remineralization depth as an additional measure of the biological pump efficiency from the model simulations. This could be helpful for cross-comparison with other modeling studies focused on the biological pump.

In Section 4.1 (line 244-251), it is discussed that cGENIE underestimates surface stratification and overestimates winter-time deep mixing due to an overly-strong AMOC in the physical circulation scheme of the model. The amount of phosphate returned to the surface is a function of deep mixing that increases organic matter production there,

and this would not be modeled well if the model underestimated surface stratification. Uncertainties in the warming scenario results should be discussed.

Technical comments In line 310, "measured" should be "simulated" or "modeled"

In Figure 2, the processes shown are not correct. Microbial (heterotrophic bacterial) respiration is also part of heterotrophic respiration (heterotrophic respiration = zooplankton respiration + bacterial respiration) and also occurs in the euphotic zone. The current schematic makes it look like microbial respiration is a separate process from heterotrophic respiration.

In Figure 12, Data "from" Mouw et al. 2016a.

In Figure 13, Please consider putting a global uniform value for POC transport efficiency in CB next to the CBRU plot, instead of presenting the stand-alone CB plot.

In all Figures, increase font size for better legibility as figure quality is currently poor; and use constant symbols in vertical profile figures for data, CB, and CBRU comparison.

In Table 1, "eqn" to "equation" here and throughout the manuscript, and it is better to say what the difference in each reference is in the first column rather than simply attaching references.

In Table 2, "x" to checkmark or "v" as it seems to indicate that the thing with "x" mark was not included in the study. "Standard model" to "non-temperature dependent model".

In Table 3, "variable" in the first column to "parameter"

––––––––––––––––––––––––

---

## Author Response (AR1)

List of changes applied to the manuscript

"Calibration of key temperature-dependent ocean microbial processes in the cGENIE.muffin (v0.9.123) Earth system model" by K.A.Crichton et al. for GMD.

We have:

Added the model version to the title of the manuscript

Added a new temperature-dependent DOM cycling option for cGENIE, and showed its impact on our main results, to address the comments from reviewer 1

Added a clarification on ocean circulation, with a caveat that we do not test any "extreme" warming scenarios in this paper. We have also adjusted the conclusion to clarify that we find temperature-dependent POM has offset (to a large) extent stratification induced nutrient limitation since the pre-industrial period (rather than stating it as a general pattern for warming).

Adjusted the equations to ensure that all have unique parameter identifiers.

Renamed the model configurations so they are more intuitive. CB becomes STND, CBRU becomes TDEP.

Provided more information on the statistical analysis, and provided the values for the fit to data for the main simulations in the text (as table 4).

We have improved all the figures, ensuring text is not too small to be legible.

We have addressed all the individual comments from the reviewers.

We have made general improvements to the text.

Response to reviewers, "Calibration of key temperature-dependent ocean microbial processes in the cGENIE.muffin Earth system model"

**Response to reviewer 1**

"Figure 2 highlights a key opportunity that has been overlooked in this study, which is that the fractional assignment of export to the DOM pool could easily be made temperature-dependent, and that would implicitly represent temperature-dependent fast recycling processes in the upper ocean. The authors should discuss why this parameter was not included in their calibration."

We have added the option to the model of a temperature-dependence to both the production and remineralization of DOM, and assess the consequences of these new processes in the analysis of marine carbon cycling response to historical warming in the revised paper.

As described in a new and extensive subsection in the Discussion (5.2), adding temperature-dependence to the decay of DOM is relatively straight-forward as to a first order, one can start by assuming this takes place analogously to the decay of POM. Hence we adopt the same calibrated activation energy, but re-tune the scaling rate constant, as described in section 5.2.

The production of DOM is more problematic as available empirical equations such of Dunne et al. [2005], requires knowledge of either primary production (integrated across the mixed layer), or Chl*a*, neither of which is explicitly calculated in the standard (non ecosystem) configuration of cGENIE. (The standard configuration of the ocean circulation model also does not calculate a mixed layer depth.) We hence extracted just the temperature sensitivity from the regression model of Dunne et al. [2005] and apply this to the partitioning of POM vs. DOM in the model. And then tune this second parameterization.

While we find that the introduction of temperature-dependent processes in DOM cycling has a much lesser impact on global export than do temperature-dependent processes directly affecting POM, we agree with the reviewer that this makes for a more complete and rounded model development.

"Schmittner et al. 2008, GBC doi:10.1029/2007GB002953 Introduced a temperature dependent remineralization parameterization to the UVic ESCM…"

We have added Kvale et al 2015 and Kvale et al 2019 to those papers discussed in the introduction. And added Loptien and Dietze 2019 and Kvale et al 2019 to the discussion noting that circulation state is important for nutrient and carbon distributions.

"P6, Equation 5: Please either rename A or make it clear on this page that the value is different than the A in Equation 4."

We have been through all the equations and terms in the entire manuscript, including the equations associated with the new DOM parameterizations, and ensure unique symbol choices for all parameters. (Capital 'A' we reserve for fractional sea-ice area only now and indeed, further clarify this by adding the subscript 'ice'.)

"P7L89: Meyer et al. (2016) prescribed several e-folding depths, which approximates, but is not the same as, temperature dependence in export"

The reviewer is correct in that Meyer et al., (2016) prescribes different e-folding depths to parameterise the potential impacts of changing surface ecosystems in geological time on remineralisation, such as increasing organism size and complexity of trophic interations. Here we are

specifically referring to the export production scheme used that is the temperature-dependent scheme described by Monteiro et al., (2012). We have edited this sentence to better clarify and indeed now substituted the Monteiro et al. (2012) reference.

"P9L250: the North Pacific subsurface temperature profile is over-estimated in GENIE according to Fig 4, please correct this sentence"
Corrected in the text.

"P9L256: What are the lowest RMSE for CB and CBRU?"
We considered both the surface and full water column distributions to select the best-fit option for PO4: the (centred) RMSE for surface layer CB is 0.1700, CBRU is 0.1820; for whole ocean CB is 0.2208, CBRU is 0.2043 all in µmol/l. For O2 we use the whole ocean and depth layer 4 (283m to 411m), the (centred) RMSE depth layer 4 CB is 0.8145, CBRU is 0.8443; for whole ocean CB is 0.5476, CBRU is 0.5501 all in µmol/l. These values are represented in figure 5 and have been added to the main text for the STND (formerly CB) and TDEP (formerly CBRU) model and the new TDEP+TDOM configuration as table 4.

"P9L268: My understanding is the models are also tuned to O2 (from P9L253-256)"
Yes, this is correct. Added to the sentence in parentheses for clarity.

"P12L375: what is the Eastern Tropical North Pacific?"
This is a typo and should read "Eastern Tropical Pacific" – now corrected in the text.

"P13L380-381: DOM cycling is also changed"
We now explicitly address and discuss (in Section 5.2) how the DOM cycle is impacted by historical temperature rise, including now also accounting for temperature-dependent processes in DOM cycling. We also add 2 additional figures to the Appendix A (Fig A1 and A2) summarising the cycling of DOM, both in the present-day state, and the response to historical warming (as an anomaly), for all the main permutations of temperature-dependent parameterisations. We have noted that DOM may also be affecting NPP in tropical waters in section 5.1.

"P13 Summary: Please see Kvale et al. 2019 BG /10.5194/bg-16-1019-2019"
Kvale et al 2019 use a transient cold to warm simulation, the warm climate with over 1200ppm atmospheric CO2 and state that in the warm climate more phosphate is stored in the deep ocean due to longer residence time of deep waters. This scenario is entirely different to that used here where the transient simulation follows CO2 trajectory from 1700 CE to the present, with a max ~400ppm. At very high CO2 (such as 1200ppm) the AMOC (Atlantic meridional overturning circulation) is likely very much reduced or collapsed and this greatly reduces the return of deeper waters to the surface compared to a the situation with a strong AMOC at 400ppm (such as the present day). We have added the following to the main text in the discussion of the model response to historical warming:

"We note that circulation states and upwelling/downwelling changes can also have an impact on the distribution of carbon, oxygen and nutrients between the surface and the deep (Kvale et al 2019, Loptien and Dietze 2019), and are also model-dependent. Circulation changes are small between the pre-industrial and the present-day, unlike in the simulations in Kvale et al (2019) where very high $CO_2$ (up to 1200ppm) and high surface temperature results in large circulation pattern changes; increased nutrient storage in the deep ocean is

due to longer residence time of deep ocean water in that study (see Chikamoto et al. 2008 for the effect of Atlantic Overturning Circulation shutdown in cGENIE). In our study we have found that the temperature dependent biological pump offsets some of the effects of physical ocean response to warming (in increase near-surface nutrient recycling, so offsetting to some extent the effect of increased ocean stratification that otherwise reduces surface nutrients in the STND simulation). However, this is not to suggest that a temperature-dependent biological pump could offset the effect of extreme changes in circulation, such as an AMOC shutdown, or for far more extreme warming scenarios than that applied here. We do not test such scenarios here."

"Table 2 can be cut at no detriment to the manuscript. 'CB' and 'CBRU' are defined too late in the manuscript and the naming includes an extra (confusing) reference to BIOGEM. I suggest omitting the Table and renaming the simulations to something more descriptive, like 'Temp' and 'NoTemp', since all models contain 'Remineralization' and 'Uptake'."

We have retained Table 2 because it is now expanded with an additional alternative model configuration, now that we are also addressing DOM-linked temperature-dependent processes. However, we have noted the lack of intuitiveness in the 'CB' and 'CBRU' naming and have replaced these throughout the manuscript with the more intuitive "STND" and "TDEP" references. We have also added these references to the relevant parts in Section 2 that describe the standard and temperature-dependent model formulations.

"Figure 1 can be cut at no detriment to the manuscript"

This figure lays out the basic functioning of the biological pump, and visually summarises the two components of temperature dependence (in both nutrient uptake and remineralisation), as well as the partitioning of organic matter into POM and DOM – we think it is a useful overview of the components discussed in the text.

"Figure 2 caption: is "mixed player plankton" supposed to read "mixed phytoplankton"? Please add a key to clarify what dashed/solid/thick/thin lines, and shading, represent. Why is burial shown if there are no sediments? Should "nutrients" be PO4 (only PO4 in this model)? Why are autotrophic respiration/heterotrophic respiration/consumers shown if they are not included in the model?"

We have improved the caption to figure 2 based on these comments:

"Figure 2. Schematic of biological pump processes showing where cGENIEs export production operates. In the export production model, no mechanistic consideration of the effects of temperature within the mixed-layer (i.e. GPP vs NPP vs community production) can be considered, but heterotrophic respiration (as remineralisation) vs community production (as export production) can be considered, as well as nutrient recycling. In this study we apply temperature dependency to Organic Matter production and remineralisation that drives the biological carbon pump. We do not model burial in this version of cGENIE (but it is here for completeness). In this cGENIE configuration, the nutrient is phosphate. Dashed line indicates the cycling of nutrient (and re-supply due to circulation). Solid lines indicates the cycling of carbon."

"Figure 8. I see why the figure is normalized (the point made on P9L275), but normalization is misleading (small differences of low concentrations appear to be significant). The figure would be more informative presented without normalization, but the above point can still be made in the text."

Figure 8 is now plotted also as absolute differences, not only as normalised differences, but with the normalised difference plot retained as it is directly referenced in the text and shows that largest

proportional changes (where an anomaly alone would tend to reduce the significance of changes in ocean regions that started with low nutrient concentrations that actually see the largest proportional changes). Nutrients limitation is important in the gyres, so any increase in concentration there would likely have a large impact on primary production.

All other technical corrections suggested by reviewer 1 have been applied to the text.

**Response to reviewer 2**

"At Ea(1) of >54.5 kJ/mol POC export becomes lower compared to the non-temperature dependent model. This could be concerning given that a small variation of the Ea(1) value (0.5 kJ/mol) results in completely opposite patterns compared to the findings of the study, i.e., the temperature-dependent model simulates lower POC export than the non-temperature dependent model. The uncertainty related to this finding does not come across clearly and should be discussed. Also, it does not seem necessary for circles to be color-coded to reflect different rPOM in Figure 11."

We thank the reviewer for highlighting that this is not clear. Partly this may have been due to a missing minus sign in the exponent of Equation 5 which has now been corrected. The relationship in Figure 11 of the manuscript is not immediately obvious as temperature is effectively fixed in the calibration runs. The reviewer is correct in noting a larger activation energy (Ea1) results a greater sensitivity of remineralisation rates to temperature (Figure 1, bottom panel). However, at a constant temperature, the remineralisation rate decreases as Ea1 increases (Figure 1 here, top panel). In our calibration runs (Figure 11 in the manuscript), atmospheric $CO_2$ is restored to 278 ppm such that climate and SSTs are invariant, i.e., temperature is constant across the calibration runs. Therefore, the increased activation energies tested lead to a decrease in remineralisation rate globally. This leads to deeper remineralisation of organic matter which drives the decrease in export production (e.g., Kwon et al., 2009, Nature Geoscience), but is still related to a greater sensitivity to temperature changes (Figure 1). We have added a brief explanation of this to the text describing Figure 11 in the manuscript as well as added the plates in figure 1 (here) to figure 11 in the manuscript to clarify this point.

The main finding of the study is that warming results in a drop in POC export for STND (-2.9%), but a rather smaller change in POC export for TDEP (at -0.3% for the best-fit setting). Even with Ea(1) at >55kJ/mol this pattern still holds because POC export is stimulated on warming in TDEP due to the increased nutrient recycling, now that remineralisation is temperature-dependent. Figure 11a shows only the global mean POC export at a fixed temperature, not how that export changes on warming. For higher Ea(1) that nutrient recycling increase on warming may be slightly lower, but certainly not such that the findings of the paper are inversed. Even with Ea(1) at 60 kJ/mol (Vmax =1, POC frac = 0.008), the drop in POC export for the historical period simulation is 0.43% (compared to the drop of 0.3% for TDEP best fit). The value of 54kJ/mol was selected as it shows a better fit to $PO_4$ and $O_2$ data (fig 5 main text) than other values.

[Figure]

**Figure 1.** Top panel: The remineralisation rate calculated for with different activation energies for a constant temperature (273 K). Bottom panel: The $Q_{10}$ of the remineralisation rate, i.e., the proportional change in the remineralisation rate for an increase in 10 K.

The colour coding on figure 11 reflects the colours used in figure 5, this has been added to the figure caption.

"In Introduction, in relation to "A deeper mean remineralization depth equates to a more "efficient" biological carbon pump" it would be good to calculate the remineralization depth as an additional measure of the biological pump efficiency from the model simulations. This could be helpful for cross-comparison with other modeling studies focused on the biological pump."

This is a useful suggestion from the reviewer. The key limitation of calculating a mean remineralisation depth here is that we are calibrating the model to observations such that the resulting values for STD and TDEP should be similar. We also note that the absolute mean remineralisation depth has been shown to be strongly dependent on ocean circulation (in particular the deep water formation in the North Atlantic) (Figure 6b in Kriest et al., 2020; Biogeosciences). As such, this is likely to be model dependent.

We note that a global value does not well represent the underlying characteristic identified in this study, where the remineralisation depth is entirely dependent on local conditions. In cold high latitudes the mean remineralisation depth is far deeper than in warm low latitudes. So, the resultant global mean will be somewhere in the middle. What then happens to the carbon in the deeper ocean is a function of ocean circulation and possible burial, so is heavily model-dependent. However, we have added this information to the main text in section 4.4 [627m for the standard model (formerly called CB) and 378m ± 236m for the T dependent model (formerly called CBRU)], and the

change in the global mean depth in section 5.1 for the historical period simulation [a shallowing of 16m for T-dependent model (formerly named CBRU)]. We have also added this information in section 5.2 for the additional DOM T-dependent model (the present-day is 399m ± 255m, and a shallowing of 16m since the year 1700).

"In Section 4.1 (line 244-251), it is discussed that cGENIE underestimates surface stratification and overestimates winter-time deep mixing due to an overly-strong AMOC in the physical circulation scheme of the model. The amount of phosphate returned to the surface is a function of deep mixing that increases organic matter production there, and this would not be modeled well if the model underestimated surface stratification. Uncertainties in the warming scenario results should be discussed."

We have added a paragraph in the discussion (in response to reviewer 1), that circulation plays a role in nutrient distribution as well, and that circulation is model dependent. We further note that cGENIEs reduction in export production (driven by an increase in stratification) through the pre-industrial to present warming agrees with the current state of the art in more complex models (CMIP5 models).

The winter time mixing overestimation was only for the North Pacific region (as indicated by the temperature profile there) not for the global ocean, and is not described as "due" to an overly-strong AMOC (which was only suggested as related to the N.Atlantic temperature profile). We do not do an extensive analysis of cGENIEs circulation in this study.

However, as we model a similar pattern of reduced export production in the standard model (due to increased stratification) that is shown in the mean of CMIP5 models, and we can fairly well model $PO_4$ and $O_2$ distributions in the present day we are reasonably confident that we have captured the large scale changes in circulation for the warming scenario. The warming scenario (the historical warming) compares the results from the different model configurations, which are all subject to the same changes in circulation that may be driven by that warming. Therefore we conclude that any differences in carbon cycle parameters are due to the temperature dependent biological processes, not circulation.

In Figure 2, the processes shown are not correct. Microbial (heterotrophic bacterial) respiration is also part of heterotrophic respiration (heterotrophic respiration = zooplankton respiration + bacterial respiration) and also occurs in the euphotic zone. The current schematic makes it look like microbial respiration is a separate process from heterotrophic respiration.

Reviewer 2 is correct. This separation between surface food web (heterotrophic respiration) and sub surface processes (microbial respiration) comes from the way they are often treated in models. Here surface food webs are modelled as being separate from sub surface processes like remineralisation, whereas actually this is not strictly the case in fact. The figure has been changed so that microbial respiration also reads "heterotrophic respiration". The aim of the figure is to demonstrate where cGENIEs export production model "sits" and hopefully makes the comparison to other models' treatments of the biological pump more straight-forward.

"In Figure 13, Please consider putting a global uniform value for POC transport efficiency in CB next to the CBRU plot, instead of presenting the stand-alone CB plot."

This value has been added to the caption for CB, but we keep the figure to really emphasise that including the temperature dependence makes a large difference to ocean carbon cycling.

"In all Figures, increase font size for better legibility as figure quality is currently poor; and use constant symbols in vertical profile figures for data, CB, and CBRU comparison."
Figure quality and legibility has been improved in the revised version for all figures.

All other technical comments from reviewer 2 have been included in the revised version.

**Editor comment response**

The title has been changed to include the model name and version:

Calibration of key temperature-dependent ocean microbial processes in the cGENIE.muffin (v0.9.13) Earth system model

We thank all reviewers for their comments.

K.A.Crichton on behalf of all authors.

**Calibration of  temperature-dependent ocean microbial processes in the cGENIE.muffin (v0.9.13) Earth system model**

Katherine A. Crichton[1*], Jamie D. Wilson[2], Andy Ridgwell[3], Paul N. Pearson[1].

[1] School of Earth and Ocean Sciences, Cardiff University, UK
[2] BRIDGE, School of Geographical Sciences, University of Bristol, Bristol, UK
[3] Department of Earth and Planetary Sciences, University of California, Riverside, CA 92521, USA
* now at School of Geography, University of Exeter, EX4 4RJ, UK

*Correspondence to*: Katherine A. Crichton (k.a.crichton@exeter.ac.uk)

**Abstract.** Temperature is a master parameter in the marine carbon cycle, exerting a critical control on the rate of biological transformation of a variety of solid and dissolved reactants and substrates. Although in the construction of numerical models of marine carbon cycling, temperature has been long-recognised as a key parameter in the production and export of organic matter at the ocean surface, its role in the ocean interior is much less frequently accounted for. There, bacteria (primarily) transform sinking particulate organic matter (POM) into its dissolved constituents and  consume dissolved oxygen (and/or other electron acceptors such as sulphate). The  nutrients and carbon thereby released then become available for transport back to the surface, influencing biological productivity and atmospheric $p$CO$_2$, respectively. Given the substantial changes in ocean temperature occurring in the past, as well as in light of current anthropogenic warming, appropriately accounting for the role of temperature in marine carbon cycling may be critical to correctly projecting changes in ocean deoxygenation as well as the strength of feedbacks on atmospheric $p$CO$_2$.

Here we  extend and calibrate a temperature-dependent representation of marine carbon cycling in the cGENIE.muffin Earth system model, intended for both past and future climate applications. In this, we combine a temperature-dependent remineralisation scheme for sinking organic matter with a biological export production scheme that also includes a dependence on ambient seawater temperature. Via a parameter ensemble, we jointly calibrate the two parameterisations by statistically contrasting model projected fields of nutrients, oxygen, and the stable carbon isotopic signature ($\delta^{13}$C) of dissolved inorganic carbon in the ocean, with modern observations. We additionally explore the role of temperature in the creation and recycling of dissolved organic matter (DOM) and hence its impact on global carbon cycle dynamics.

We find that for the present-day, the temperature-dependent version shows as-good-as or better fit to data than the existing tuned non-temperature-dependent version of the cGENIE.muffin. The main impact of accounting for temperature-dependent remineralisation of POM is in driving higher rates of remineralisation in warmer waters, in turn driving  a more rapid return of nutrients to the surface and thereby,  stimulating organic matter production. As a result, more POM is exported below 80m but on average reaches shallower depths in mid and low latitude warmer waters, as compared to the standard

model. Conversely, at higher latitudes, colder water temperature reduces the rate of nutrient resupply to the surface and POM reaches greater depth on average as a result of slower in-situsub-surface rates of remineralisation and POM reaches greater depth on average. Further adding temperature-dependent DOM processes changes this overall picture only a little, with a slight weakening of export production at higher latitudes.

We As an illustrativeon application of the new model configuration and calibration, we take the example of historical warming alsoand briefly assess the implications for global carbon cycling of accounting for a more complete set of temperature-dependent processes in the ocean, for which of including a more complete set of temperature-dependent parameterisations by analysing a series of historical transient experimentsunder historical warming. We find that between the pre-industrial (ca. 1700) and the present day(year 2010), in response to a simulated air temperature increase of 0.9°C and an associated projected mean ocean warming of 0.12°C (0.6°C in surface waters and 0.02°C in deep waters), a reduction in particulate organic carbon (POC) export at 80m of just 0.3% occurs (or 0.7%xxx including a temperature-dependent DOM response). However, due to this increased recycling nearer the surface, the efficiency of the transfer of carbon away from the surface (at 80m) to the deep ocean (at 1040m) is reduced by 5%. In contrast, with no assumed temperature-dependent biological processes impacting any of the production or remineraliszation of either POM or DOM, global POC export at 80m falls by 2.9% between the pre-industrial and present day as a consequence of ocean stratification and reduced nutrient resupply to the surface. This suggests that increased nutrient recycling in warmer conditions offsets some of the stratification-induced surface nutrient limitation in a warmer rapidly warming world, and withthat less carbon (and nutrients) then reaches reaching the inner and deep oceanocean interior. This extension to the cGENIE.muffin Earth system model provides it with additional capabilities in addressing marine carbon cycling in warmer past and future worlds. by 5% between 80 and 1040m depth. Between 1700 and 2010 CE, we simulate a reduction of over 5% in the proportion of carbon exported at 80m that reaches 1040m depth. 
[revised manuscript text omitted]

Conceptually, the fate of DOM is relatively easy to address as in theory, it should display an analogous temperature-dependent response as per POM. In the standard configuration of the cGENIE Earth system model (e.g. as per Ridgwell et al., 2007; Cao et al., 2009), the lifetime of DOM is fixed and set at a value of 0.5 years following Najjar et al. [2007]. To explore the wider role of temperature in the marine carbon cycle including that of DOM, we also now implement,and as a further option in the model, the samewe add a similar temperature-dependence to the remineraliszation of DOM as for POM (Equation 5):

$$k(T)_{DOM} = \beta A_{DOM} e^{\left(-E_a/RT\right)} \tag{7}$$

where $E_a$ is the activation energy and is assigned a value of 54000 J mol$^{-1}$, and $R$ and $T$ are the gas constant (J K$^{-1}$ mol$^{-1}$) and absolute temperature (K), respectively. $k(T)_{DOM}$ is the rate constant (year$^{-1}$) controlling the decay of DOM and replaces the previous fixed lifetime value of 2.0 year$^{-1}$.

AssumingThe assumption that the same activation envergy applies to DOM as per for POM (that was calibtrated in conjunction with an assumed sinking rate of POM of 125 m d$^{-1}$), leads to a calibrated value of $A$ in Eq. 5 (9.0×10$^{11}$). However, implementing Eq. 7 withi.e. setting $\beta_{DOM} = 9.0 \times 10^{11}$ in Eq. 7) l$A_{DOM} = 9.0 \times 10^{11}$ leads to a global mean DOM lifetime of approximately 0.02 years, compared to the value of 0.5 years in the standard model (Ridgwell et al., 2007) that follows Najjar et al. (2007). This might: (i) reflect a different mean quality (reactivityand hence a different activation energy) ofto the organic matter assumed to constitute DOM as opposed to POM;, and/or, (ii) reflect the dispersed nature of DOM versus the more concentrated POM and/or differences in the associated bacterial biomass;, and/or (iii) that the assumed sinking speed of 125 m d$^{-1}$ is simply too unrealistically fast fast an assumption. While one could argue that (i) also may imply that a different activation energy for DOM is also required. However, to simplify and linearize 
[revised manuscript text omitted]

940